# Preventing Dimensional Collapse in Self-Supervised Learning via Orthogonality Regularization

**Junlin He**
The Hong Kong Polytechnic University
Hong Kong SAR, China
junlinspeed.he@connect.polyu.hk

**Jinxiao Du**
The Hong Kong Polytechnic University
Hong Kong SAR, China
jinxiao.du@connect.polyu.hk

**Wei Ma**[*]
The Hong Kong Polytechnic University
Hong Kong SAR, China
wei.w.ma@polyu.edu.hk

## Abstract

Self-supervised learning (SSL) has rapidly advanced in recent years, approaching the performance of its supervised counterparts through the extraction of representations from unlabeled data. However, dimensional collapse, where a few large eigenvalues dominate the eigenspace, poses a significant obstacle for SSL. When dimensional collapse occurs on features (e.g. hidden features and representations), it prevents features from representing the full information of the data; when dimensional collapse occurs on weight matrices, their filters are self-related and redundant, limiting their expressive power. Existing studies have predominantly concentrated on the dimensional collapse of representations, neglecting whether this can sufficiently prevent the dimensional collapse of the weight matrices and hidden features. To this end, we first time propose a mitigation approach employing orthogonal regularization (OR) across the encoder, targeting both convolutional and linear layers during pretraining. OR promotes orthogonality within weight matrices, thus safeguarding against the dimensional collapse of weight matrices, hidden features, and representations. Our empirical investigations demonstrate that OR significantly enhances the performance of SSL methods across diverse benchmarks, yielding consistent gains with both CNNs and Transformer-based architectures. Our code will be released at https://github.com/Umaruchain/OR_in_SSL.git.

## 1 Introduction

Self-supervised learning (SSL) has established itself as an indispensable paradigm in machine learning, motivated by the expensive costs of human annotation and the abundant quantities of unlabeled data. SSL endeavors to produce meaningful representations without the guidance of labels. Recent developments have witnessed joint-embedding SSL methods achieving, or even exceeding the supervised counterparts (Misra & Maaten 2020, Bardes et al. 2022, Caron et al. 2020, Chen, Fan, Girshick & He 2020, Chen, Kornblith, Norouzi & Hinton 2020, Chen & He 2021, Dwibedi et al. 2021, HaoChen et al. 2021, He et al. 2020, He & Ozay 2022, Jing et al. 2021, Li, Zhou, Xiong & Hoi 2020, Jing et al. 2020, Balestriero et al. 2023, Grill et al. 2020, Zbontar et al. 2021, Chen et al. 2021). The efficacy of these methods hinges on two pivotal principles: 1) the ability to learn augmentation-invariant representations, and 2) the prevention of complete collapse, where all inputs are encoded to a constant vector.

---

[*]Corresponding author.

38th Conference on Neural Information Processing Systems (NeurIPS 2024).

Efforts to forestall complete collapse have been diverse, including contrastive methods with both positive and negative pairs (He et al. 2020, Chen, Kornblith, Norouzi & Hinton 2020, Chen et al. 2021) and non-contrastive methods utilizing techniques such as self-distillation (Caron et al. 2021, Grill et al. 2020, Chen & He 2021), clustering (Caron et al. 2018, 2020, Pang et al. 2022) and feature whitening (Bardes et al. 2022, Zbontar et al. 2021, Weng et al. 2022, 2023). Notwithstanding, these methods are prone to dimensional collapse, a phenomenon where **a few large eigenvalues dominate the eigenspace**. Dimensional collapse can occur on both features (e.g. hidden features and representations) and weight matrices.

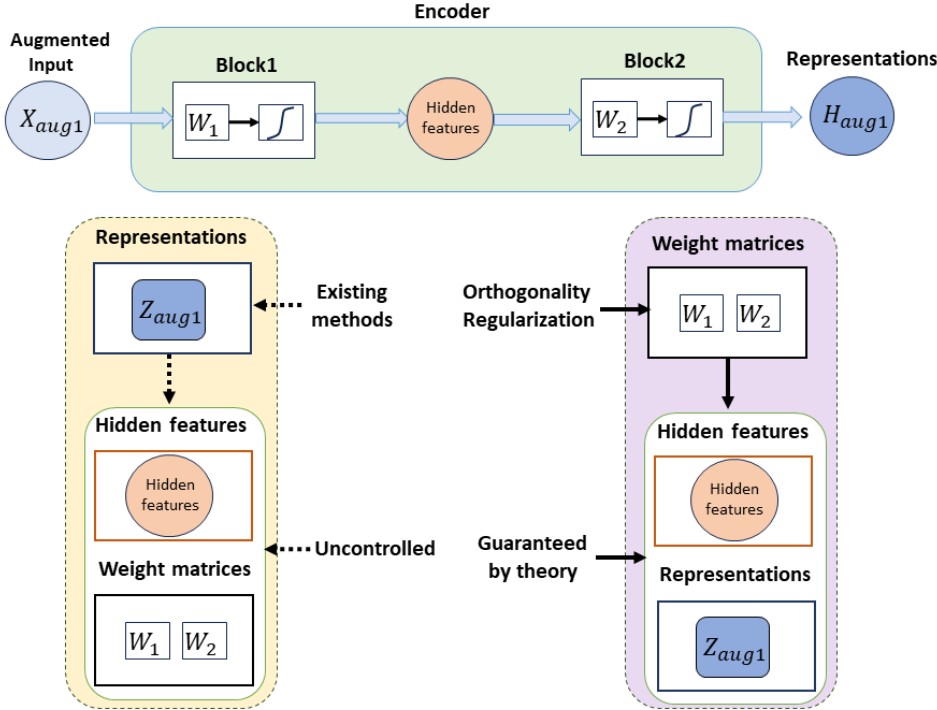

Figure 1: Illustration of dimensional collapse in SSL. We use one augmented input $X_{aug1}$ as an example: we assume that the encoder contains two basic blocks, each containing a linear operation (e.g., a linear layer or convolutional layer) and an activation function. Dimensional collapse can occur in weight matrices ($W_1, W_2$), hidden features, and the finally obtained representations. Existing methods act directly on representations and expect to affect hidden features and weight matrices indirectly, which has no guarantee in theory; our method directly constrains weight matrices and indirectly influences hidden features and representations, which can be guaranteed by theoretical analysis.

To prevent dimensional collapse of representations, as depicted in Figure 1, existing methods include modifying representations in downstream tasks (He & Ozay 2022), whitening representations directly (i.e. removing the projector) (Jing et al. 2021), incorporating regularizers on representations during pretraining (Huang et al. 2024, Hua et al. 2021). However, whether they sufficiently prevent the dimensional collapse of weight matrices and hidden features remains unknown (i.e., no theoretical guarantee) (Pasand et al. 2024). In Appendix A.1, we further demonstrate that whitening representations directly to eliminate the dimensional collapse of representations cannot adequately remove the dimensional collapse of weight matrices.

To address these challenges, we first time propose a mitigation approach employing orthogonal regularization (OR) across the encoder, targeting both convolutional and linear layers during pretraining. It is natural that OR prevents the dimensional collapse of weight matrices as it ensures weight matrices orthogonality, keeps the correlation between its filters as low as possible, and lets each filter have a norm of 1. For features (e.g. hidden features and representations), orthogonal weight matrices can promote uniform eigenvalue distributions and thus prevent the domination of eigenspaces by a

limited number of large eigenvalues, as theoretically substantiated by Huang et al. (2018), Yoshida & Miyato (2017), Rodríguez et al. (2016).

In our study, we introduce and assess the effects of two leading orthogonality regularizers, Soft Orthogonality (SO) and Spectral Restricted Isometry Property Regularization (SRIP), on SSL methods. We examine their integration with 13 modern SSL methods from Solo-learn and LightSSL, spanning both contrastive and non-contrastive methods (Chen, Fan, Girshick & He 2020, Chen et al. 2021, Grill et al. 2020, Caron et al. 2021, Dwibedi et al. 2021). Our findings indicate a consistent enhancement in linear probe accuracy on CIFAR-100 using both CNNs and Transformer-based architectures and OR exhibits a good scaling law at the model scale. Furthermore, when applied to BYOL trained on IMAGENET-1k, OR significantly improves the downstream performance on both classification and object detection tasks, suggesting its applicability to large-scale SSL settings. Remarkably, OR achieves these enhancements without necessitating modifications to existing SSL architectures or hyperparameters.

In summary, we present three major contributions:

- We systematically study the phenomenon of dimensional collapse in SSL, including how feature whitening and network depth affect the dimensional collapse of weight matrices and hidden features.
- We first time introduce orthogonal regularization (OR) as a solution to prevent the dimensional collapse of weight matrices, hidden features, and representations during SSL pretraining.
- Our extensive experimental analysis demonstrates OR's substantial role in enhancing the performance of state-of-the-art joint-embedding SSL methods with a wide spectrum of backbones.

## 2 Related Work

### 2.1 Self-Supervised Learning

Self-supervised Learning (SSL) aims to learn meaningful representations from unlabeled data. Existing SSL methods can be broadly classified into two categories: generative and joint embedding methods. This paper concentrates on joint-embedding methods, which learn representations by aligning the embeddings of different augmented views of the same instance. Joint-embedding methods further subdivide into contrastive and non-contrastive methods. Contrastive methods, such as those proposed by He et al. (2020), Chen, Kornblith, Norouzi & Hinton (2020), Chen et al. (2021), treat each sample as a distinct class and leverage the InfoNCE loss (Oord et al. 2018) to bring representations of positive pairs closer together while distancing those of negative pairs in the feature space. These methods generally require a substantial number of negative samples for effective learning. In contrast, non-contrastive methods eschew the use of negative samples. They instead employ various techniques such as self-distillation (Caron et al. 2021, Grill et al. 2020, Chen & He 2021), clustering (Caron et al. 2018, 2020, Pang et al. 2022) and feature whitening (Bardes et al. 2022, Zbontar et al. 2021, Weng et al. 2022, 2023). Our empirical findings indicate that incorporating OR enhances the performance of both contrastive and non-contrastive SSL methods. The exploration of its effects on generative methods remains for future work.

### 2.2 Dimensional Collapse in SSL

Dimensional collapse plagues both generative and joint embedding SSL methods (Zhang et al. 2022, Jing et al. 2021, Zhang et al. 2021, Tian et al. 2021). To prevent the dimensional collapse of representations, existing work has typically focused on imposing constraints on the covariance matrix of the representations, including modifying representations in downstream tasks (He & Ozay 2022), removing the projector (Jing et al. 2021), incorporating regularizers on representations during pretraining (Huang et al. 2024, Hua et al. 2021). However, these strategies face challenges such as performance degradation upon removing the projector, not addressing collapse during pre-training, and failing to prevent dimensional collapse in hidden features and weight matrices within the encoder (referred to Appendix A.1). This motivates us to regularize the weight matrices of the DNNs directly in SSL.

## 2.3 Orthogonality Regularization

Orthonormality regularization, which is applied in linear transformations, can improve the generalization and training stability of DNNs (Xie et al. 2017, Huang et al. 2018, Saxe et al. 2013). OR has demonstrated its effects on tasks including supervised/semi-supervised image classification, image retrieval, unsupervised inpainting, image generation, and adversarial training (Bansal et al. 2018, Balestriero et al. 2018, Balestriero & Baraniuk 2020, Xie et al. 2017, Huang et al. 2018). Efforts to utilize orthogonality in network training have included penalizing the deviation of the gram matrix of each weight matrice from the identity matrix (Xie et al. 2017, Bansal et al. 2018, Balestriero et al. 2018, Kim & Yun 2022) and employing orthogonal initialization (Xie et al. 2017, Saxe et al. 2013). For more stringent norm preservation, some studies transform the convolutional layer into a doubly block-Toeplitz (DBT) matrix and enforce orthogonality (Qi et al. 2020, Wang et al. 2020).

In this work, we first time investigate the efficacy of two orthogonality regularizers, Soft Orthogonality (SO) and Spectral Restricted Isometry Property (SRIP) in SSL (Bansal et al. 2018). These regularizers aim to minimize the distance between the gram matrix of each weight matrix and the identity matrix—measured in Frobenius and spectral norms, respectively.

## 3 Preliminaries

### 3.1 Settings of Self-supervised Learning

In this section, we present the general settings for joint-embedding SSL methods. We consider a large unlabelled dataset $X \in \mathbb{R}^{N \times D}$, comprising $N$ samples each of dimensionality $D$. The objective of SSL methods is to construct an effective encoder $f$ that transforms raw data into meaningful representations $Z = f(X)$, where $Z \in \mathbb{R}^{N \times M}$ and $M$ denotes the representation dimensionality. The learning process of SSL methods is visually represented in Figure 2, where data augmentations transform $X$ into two augmented views $X_{\text{aug1}}, X_{\text{aug2}} \in \mathbb{R}^{D \times N}$. A typical joint-embedding SSL architecture encompasses an encoder $f$ and a projector $p$. These components yield encoder features $Z_{\text{aug1}} = f(X_{\text{aug1}})$ and $Z_{\text{aug2}} = f(X_{\text{aug2}})$, as well as projection features $H_{\text{aug1}} = p(Z_{\text{aug1}})$and $H_{\text{aug2}} = p(Z_{\text{aug2}})$. During training, the parameters of $f$ and $p$ are optimized via backpropagation to minimize the discrepancy between $H_{\text{aug1}}$ and $H_{\text{aug2}}$. To prevent the encoder $f$ from producing a constant feature vector, contrastive methods utilize negative samples, and non-contrastive methods employ strategies such as the self-distillation technique.

The efficacy of the encoder $f$ is usually assessed by the performance of the $C$-class classification as downstream tasks. Specifically, given a labeled dataset containing samples $X_s \in \mathbb{R}^{S \times D}$ and their corresponding labels $Y_s \in \mathbb{R}^{S \times C}$, where $S$ is the sample number. Then, a linear layer $g$ parameterized by $W_c \in \mathbb{R}^{C \times M}$ is appended on top of the learned representations $Z_s = f(X_s)$, and thus the classification task can be fulfilled by minimizing the cross-entropy between $\texttt{softmax}(g(Z_s))$ and $Y_s$. There are two strategies for the fine-tuning: 1) non-linear fine-tuning, which trains both $g$ and $f$ in the downstream tasks, and 2) linear evaluation, which freezes $f$ and only trains $g$ (referred to as the linear probe).

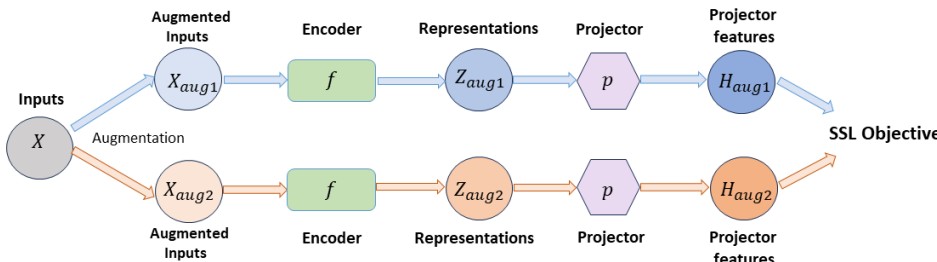

Figure 2: Illustration of joint-embedding SSL methods. This is a general structure. Different augmented inputs can be passed either by shared weight Encoder and Projector or by independent Encoder and Projector, depending on different SSL methods.

## 3.2 Orthogonality Regularizers

We introduce two orthogonality regularizers: Soft Orthogonality (SO) and Spectral Restricted Isometry Property Regularization (SRIP), which are seamlessly integrable with linear and convolutional layers.

Consider a weight matrix $W \in \mathbb{R}^{input \times output}$ in a linear layer, where $input$ and $output$ denote the number of input and output features, respectively. In line with Bansal et al. (2018), Xie et al. (2017), Huang et al. (2018), we reshape the convolutional filter to a two-dimensional weight matrix $W \in \mathbb{R}^{input \times output}$, while we still use the same notation $W$ for consistency. To be specific, $input = S \times H \times C_{in}$ and $output = C_{out}$, with $C_{in}$ and $C_{out}$ being the number of input and output channels, and $S$ and $H$ representing the width and height of the filter, respectively.

The SO regularizer encourages the weight matrix $W$ to approximate orthogonality by minimizing the distance between its Gram matrix and the identity matrix. This is quantified by the Frobenius norm as follows:

$$\text{SO}(W) = \begin{cases} \left\| W^T W - I \right\|_F^2, & \text{if } input > output, \\ \\ \left\| W W^T - I \right\|_F^2, & \text{otherwise}, \end{cases} \tag{1}$$

where $I$ is the identity matrix of appropriate size.

The SRIP regularizer employs the spectral norm to measure the deviation from orthogonality, which is defined as:

$$\text{SRIP}(W) = \begin{cases} \sigma(W^T W - I), & \text{if } input > output, \\ \\ \sigma(W W^T - I), & \text{otherwise}. \end{cases} \tag{2}$$

where $\sigma(\cdot)$ denotes the spectral norm operator. Due to the high computational cost posed by the spectral norm, the power iteration method (Yoshida & Miyato 2017, Bansal et al. 2018) with two iterations is used for the estimation. The process for estimating $\sigma(W^T W - I)$ is:

$$u = (W^T W - I)v, \quad v = (W^T W - I)u, \quad \sigma(W^T W - I) = \frac{\|v\|_2}{\|u\|_2}, \tag{3}$$

where $v \in \mathbb{R}^{input}$ is a vector initialized randomly from a normal distribution.

## 4 Incorporating OR into SSL

This section details the integration of OR with SSL methods. To be specific, we employ OR across the encoder, targeting both convolutional and linear layers during pretraining. We represent the SSL method's loss function as $Loss_{SSL}$. Our overall optimization objective is the minimization of the combined loss equation:

$$Loss = Loss_{SSL} + \gamma \cdot Loss_{OR}, \tag{4}$$

where $Loss_{OR}$ is defined as $\sum_{W \in f} SO(W)$ or $\sum_{W \in f} SRIP(W)$, depending on the selected orthogonality regularizers. The term $\gamma$ serves as a hyperparameter that balances the SSL objective and OR loss. Notably, we only perform OR on the weight matrices located within the linear and convolutional layers of the encoder $f$. OR provides a versatile regularization strategy for the encoder $f$, facilitating its application across various SSL methods without necessitating modifications to the network designs or existing training protocols.

## 5 Analysis of Dimensional Collapse and the Effects of OR in SSL

In this section, we show that dimensional collapse happens not only to the representations (i.e. output of the encoder), but also to weight matrices and hidden features of the encoder. We also compare the feature whitening technique used by one previous method, VICREG (Bardes et al. 2022), with OR. Migrating to BYOL, we find that the feature whitening technique only solves the dimensional collapse at the feature level, but instead accelerates the collapse of the weight matrices, and it even leads to lower performance of the downstream tasks as shown in Table 1. In contrast, OR can

eliminate the dimensional collapse of weight matrices and thus the dimensional collapse of hidden features and representations.

In Appendix A.1, we further reveal that the original VICREG has a dimensional collapse problem with its weight matrices, which could not be solved by removing its projector. Adding OR to VICREG eliminates this problem and boosts the performance.

Table 1: Comparison of the feature whitening technique from VICREG and SO on CIFAR-10.

| Methods | BOYL | BYOL with SO | BYOL with feature whitening |
|---------|------|--------------|------------------------------|
| Top-1 | 92.92 | **93.04** | 92.66 |
| Top-5 | 99.83 | **99.84** | 99.79 |

To study the eigenspace of a matrix, we utilize the normalized eigenvalues defined in He & Ozay (2022):

**Definition 1 (Normalized eigenvalues)** *Given a specific matrix $T \in \mathbb{R}^{N \times D}$, where $N$ is the sample number and $D$ is the feature dimension, we first obtain its covariance matrix $\Sigma_T \in \mathbb{R}^{D \times D}$. Then we perform an eigenvalue decomposition on $\Sigma_T$ to obtain its eigenvalues $\{\lambda_i^T\}_{i=1}^D = \{\lambda_1^T, \cdots, \lambda_i^T, \cdots, \lambda_D^T\}$ in descending order. And we obtain the normalized eigenvalues by dividing all eigenvalues by the max eigenvalue $\lambda_1^T$, denoted as $\{\lambda_1^T/\lambda_1^T, \cdots, \lambda_i^T/\lambda_1^T, \cdots, \lambda_D^T/\lambda_1^T\}$. To simplify the denotation, we reuse $\{\lambda_i^T\}_{i=1}^D$ to denote normalized eigenvalues of $T$.*

These normalized eigenvalues are less than or equal to 1, where a larger value in one dimension indicates more information contained, and vice versa. We argue that if normalized eigenvalues drop very quickly, this means that only a few dimensions in the eigenspace contain meaningful information and also means that dimensional collapse has occurred.

We train three BYOL (Grill et al. 2020) models, without OR, with the feature whitening technique (e.g. Variance and Covariance regularization) from VICREG and with OR, respectively. We choose randomly initialized ResNet18 as the backbone (i.e. encoder) and train the three models on CIFAR-10 for 1,000 epochs, following the same recipe of Da Costa et al. (2022). Importantly, SO is selected as the orthogonality regularize, and $\gamma$ is set to $1e-6$. For the feature whitening technique, we impose the Variance and Covariance regularization from VICREG on the output of the predictor in BYOL as two additional loss terms, the former to ensure the informativeness of individual dimensions and the latter to reduce the correlation between dimensions. Following the solo-learn settings, we set the two loss term hyperparameters to $\gamma_{vic}$ and $\gamma_{vic} * 0.004$, and then tune the $\gamma_{vic}$ from $1e-3$ to $1e-5$.

After training, we calculate the normalized eigenvalues of both weight matrices and features (e.g. input features, hidden features, representations). ResNet18 contains four basic blocks, each containing four convolutional layers, and we visualize the normalized eigenvalues of the last convolutional layer in each block. Hidden features are the outputs of four basic blocks in ResNet18. We use the first batch in the test set as input features with batchsize 4,096 and feature dimension 3072 (32*32*3). Results are analyzed in the following sections.

### 5.1 Dimensional Collapse of Weight Matrices

We first examine the weight matrices of the encoder. Similar to 5.1, all the weight matrices are viewed as two-dimensional matrices, and on top of them, we can calculate their normalized eigenvalues. For a specific weight matrix $W \in \mathbb{R}^{input \times output}$ in a neural network layer, we denote $\{\lambda_i^W\}_{i=i}^{output}$ as the normalized eigenvalues of this layer.

As shown in Figure 3, X and Y axis is the $i$ index and $i$-th values of $\{\lambda_i^W\}_{i=i}^{output}$, respectively. It is clear that with OR, the eigenvalues of the convolutional layers of different depths decay more slowly, which means that their filters are less redundant and more diverse. In particular, we note that the deepest convolutional layer (i.e. layer4_512) has a much faster decay rate of the eigenvalues compared to the other convolutional layers in the absence of OR. Notably, the feature whitening technique does not alleviate this phenomenon. OR could significantly improve this. OR requires the weight matrix to be as orthogonal as possible, which means that the diagonal elements of its covariance matrix are as identical as possible, and the off-diagonal elements will be close to 0. Then

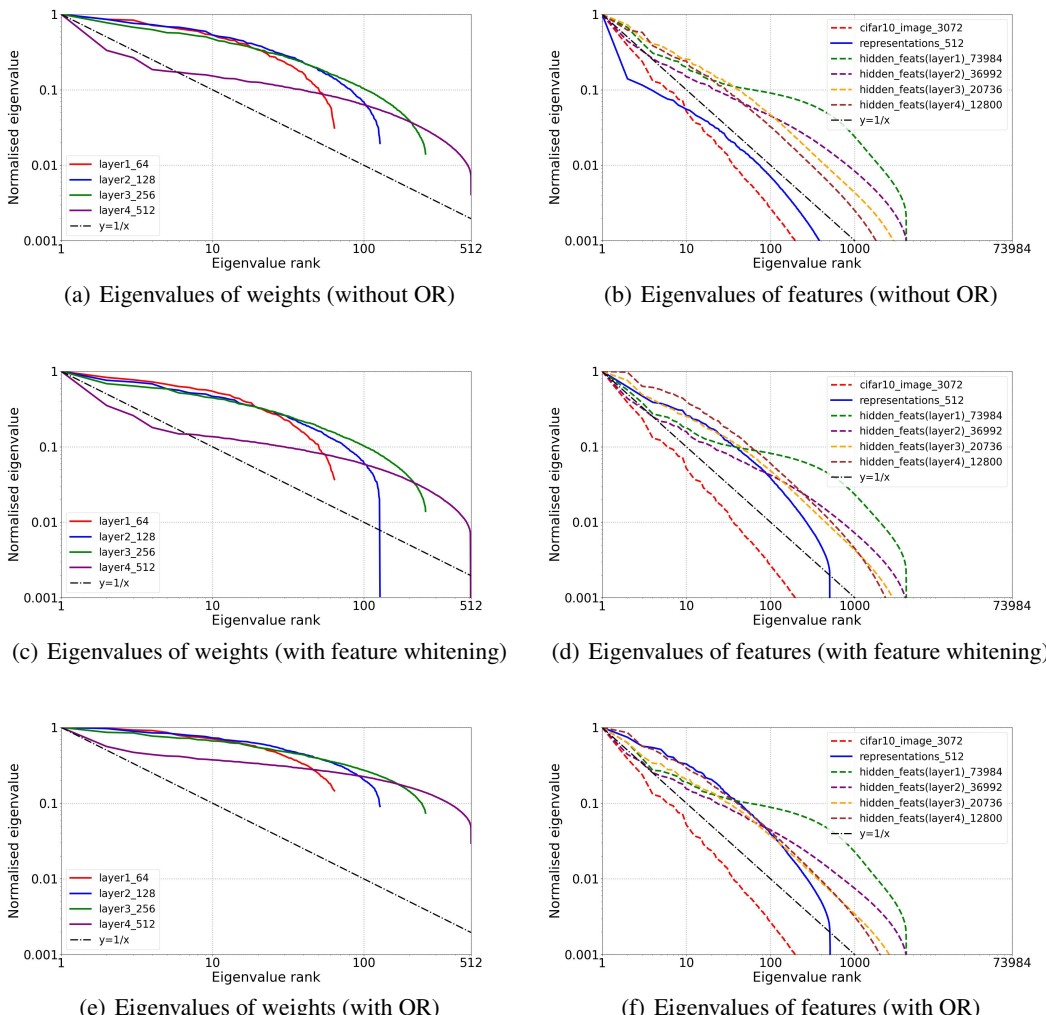

Figure 3: Eigenspectra of both weights and features within the encoder (ResNet18). The features are collected on the first batch of the test set (batchsize 4,096). We pretrain BYOL without OR, with feature whitening from VICREG, and with OR on CIFAR-10. The x-axis and y-axis are both log-scaled. The solid line represents that all eigenvalues are positive, the dashed line represents the existence of eigenvalues that are non-positive, and the number of eigenvalues is represented behind the underline.

the eigenvalue decomposition of such a covariance matrix will have elements in $\{\lambda_i^W\}_{i=i}^{output}$ that are all close to 1 (verified in Appendix A.2).

## 5.2 Dimensional Collapse of Features

As the weight matrices become orthogonal, the distribution of their outputs stabilizes, thereby preventing the dimensional collapse of hidden features and representations. This property has been demonstrated in vector form by Huang et al. (2018) and is now presented in matrix form:

**Proposition 1** *For a specific weight matrix $W \in \mathbb{R}^{input \times output}$ and $X \in \mathbb{R}^{N \times input}$, comprising $N$ samples each of dimensionality $input$. We denote $\bar{X}$ and $\bar{S}$ as the sample means of $X$ and $S$, respectively. Let $S = XW$, where $W^T W = I$. The covariance matrix of $X$ is $\Sigma_X = \frac{(X-\bar{X})^T \cdot (X-\bar{X})}{N-1}$. (1) If $\bar{X} = 0$ and $\Sigma_X = \sigma^2 I$, then $\bar{S} = 0$ and $\Sigma_S = \sigma^2 I$. (2) If $input = output$, we have $\|S\|_F^2 = \|X\|_F^2$. (3) Given the back-propagated gradient $\frac{\partial L}{\partial S}$, we have $\left\|\frac{\partial L}{\partial S}\right\|_F^2 = \left\|\frac{\partial L}{\partial X}\right\|_F^2$.*

The first point of Proposition 1 illustrates that in each layer of DNNs, the orthogonal weight matrix preserves the normalization and de-correlation of the output $S$, assuming the input is whitened. This reveals that as the network gets deeper, the hidden features of each layer and the final representations do not tend to collapse. Moreover, orthogonal filters maintain the norm of both the output and the back-propagated gradient information in DNNs, as demonstrated by the second and third points of Proposition 1.

To verify that the dimensional collapse of features can be eliminated by OR, we visualize the normalized eigenvalues of features (input features, hidden features, and representations) as shown in Figure 3. Without the OR constraint, features located in deeper layers (i.e. representations) will have the fastest eigenvalues decay rate, and the distributions of eigenvalues of hidden features vary considerably at different depths. After adding the OR, it can be seen that the decay rates of hidden features at layers 3 and 4 are almost the same, while the eigenvalues of representations decay much more slowly. We also visualize the representations in Appendix A.3, which also verifies that the dimensional collapse of the representations is mitigated. Interestingly, the effect of OR on the feature level is similar to that of the feature whitening technique, however, the latter is unable to eliminate the dimensional collapse in the weight matrices.

## 6 Numerical Experiments

We study the effects of OR on SSL methods through extensive experiments. we first demonstrate that OR improves the classification accuracy on CIFAR-10, CIFAR-100, and IMAGENET100, and the improvement is consistent across different backbones and SSL methods. On the large-scale dataset IMAGENET-1k (Deng et al. 2009), OR boosts the classification accuracy on both in-distribution and out-distribution datasets (i.e. transfer learning datasets), demonstrating consistent improvement. Moreover, OR also enhances the performance in downstream tasks(e.g. object detection).

**Baseline methods and datasets.** We evaluated the effect of adding OR to 13 modern SSL methods, including 6 methods implemented by solo-learn (MOCOv2plus, MOCOv3, DINO, NNBYOL, BYOL, VICREG)  (Chen & He 2021, Chen et al. 2021, Grill et al. 2020, Dwibedi et al. 2021, Caron et al. 2021) and 10 methods implemented by LightlySSL (BarlowTwins, BYOL, DCL, DCLW, DINO, Moco, NNCLR, SimCLR, SimSiam, SwaV)  (Zbontar et al. 2021, Yeh et al. 2022, Caron et al. 2020, Chen & He 2021). We pretrain SSL methods on CIFAR-10, CIFAR-100, IMAGENET-100 and IMAGENET-1k and evaluate transfer learning scenarios on datasets including CIFAR-100, CIFA-10 (Krizhevsky et al. 2009), Food-101 (Bossard et al. 2014), Flowers-102 (Xia et al. 2017), DTD (Sharan et al. 2014), GTSRB (Haloi 2015). We evaluate the objection detection task on PASCAL VOC2007 and VOC2012 (Everingham et al. 2010). Detailed descriptions of datasets and baseline SSL methods are shown in Appendix A.4 and A.5, respectively.

**Training and evaluation settings.**  For each SSL method, we use the original settings in solo-learn (Da Costa et al. 2022) and LightlySSL. These settings include the network structure, loss function, training policy (training epochs, optimizers, and learning rate schedulers) and data augmentation policy. The splits of the training and test set follow torchvision Marcel & Rodriguez (2010). For all the classification tasks, we report the linear probe or KNN accuracy; for the objection detection task, we perform non-linear fine-tuning. Details of training, parameter tuning, and evaluation are presented in Appendix A.6. It is worth noting that the Solo-learn and LightlySSL setups are not the same as the official implementation of the SSL methods, e.g., there is no use of multi-crop augmentation in DINO, and there is no exceptionally long training epoch. We leave experiments on migrating OR to the official implementation for future work.

**Recipe of adding OR.** For OR, $\gamma$ of SRIP is tuned from $\{1e-3, 1e-4, 1e-5\}$ and $\gamma$ of SO is tuned from $\{1e-5, 1e-6, 1e-7\}$ on a validation set. When you want to add OR to your SSL pre-training, you simply pass the encoder into the loss function, and then you just need to set $\gamma$ of the OR according to the backbone and regularizer you use as shown in Table 9 of Appendix A.6.

### 6.1   OR is Suitable for Different Backbones and SSL Methods

After pretraining on CIFRA-100, for each SSL method, we report the corresponding classification accuracy as shown in Table 2. Both two orthogonality regularizers consistently improve the linear classification accuracy. Note that OR boosts the performance of both constrastive (MoCov2plus,

Mocov3) and non-contrastive methods(BYOL, NNBYOL, DINO) as they are all susceptible to dimensional collapse (Zhang et al. 2022, Jing et al. 2021, Zhang et al. 2021, Tian et al. 2021). Non-contrastive methods gain more improvements in contrast to contrastive methods. When we use ResNet18, MOCOv3 improves 3% on Top-1 accuracy while DINO and NNBYOL improve 6% and 5%, respectively. OR can also boost the performance of the Sota method like BYOL by 1%. When we scale to ResNet 50 and WideResnet28w2, OR consistently boosts their performance. Moreover, the additional time overhead of adding OR to SSL is low compared to the original training time (referred to A.7).

Table 2: Classification accuracy on CIFAR-100 (CNN backbones). SSL methods (in Solo-learn) are trained with or without OR on CIFAR-100. The best results are in **bold**, the second best in *italics*.

| Methods | | MOCOv2plus | | MOCOv3 | | DINO | | NNBYOL | | BYOL | |
|---|---|---|---|---|---|---|---|---|---|---|---|
| Encoder | Regularizer | Top-1 | Top-5 | Top-1 | Top-5 | Top-1 | Top-5 | Top-1 | Top-5 | Top-1 | Top-5 |
| ResNet18 | - | 69.51 | 91.32 | 67.13 | 89.83 | 59.13 | 86.41 | 68.87 | 90.98 | 71.15 | 92.17 |
| | SO | **69.72** | 91.56 | *68.63* | *90.85* | *61.41* | *87.94* | *71.40* | 92.12 | **72.15** | *92.48* |
| | SRIP | *69.67* | **91.92** | **69.37** | **90.88** | **62.75** | **88.37** | **71.97** | **92.80** | *71.52* | **92.49** |
| ResNet50 | - | **73.70** | 93.16 | 66.87 | 89.47 | 52.67 | 80.68 | 72.31 | **92.94** | 74.20 | 93.44 |
| | SO | *73.40* | **93.45** | *69.22* | *90.76* | *57.30* | *84.10* | *72.87* | 92.77 | **74.57** | **93.83** |
| | SRIP | 73.32 | *93.20* | **70.30** | **90.97** | **59.93** | **86.20** | **72.97** | *92.91* | *74.39* | *93.61* |
| WideResnet28w2 | - | *61.80* | 87.69 | 57.73 | 84.82 | 53.42 | *82.62* | 64.00 | **89.49** | 60.87 | 86.96 |
| | SO | **62.09** | 87.87 | **58.86** | **85.69** | **56.07** | **84.87** | **64.50** | 89.22 | *60.96* | **87.34** |
| | SRIP | 61.37 | **87.94** | *58.79* | *85.11* | *53.93* | 83.27 | *64.23* | 89.00 | **61.10** | *87.31* |

In addition to CNN backbones, OR is also able to improve SSL performance on Transformer-based backbones (e.g., VIT) (Han et al. 2022, Dosovitskiy et al. 2020, Zhou et al. 2021). We pretrain DINO on CIFAR-100 with different depths of VITs. As shown in Table 3, with the increasing depth of the VIT, the original DINO performance is increasing, and OR is able to increase their performance even further, which exhibits a good scaling law. Interestingly, under the Transformer-based architecture, OR is able to improve performance more (up to 12%) compared to CNN backbones. This is consistent with some existing studies (RoyChowdhury et al. 2017) that linear layers are more likely to have redundant filters than convolutional layers in DNNs, i.e., more prone to dimensional collapse.

Table 3: Classification accuracy on CIFAR-100 (VITs). DINO (in Solo-learn) is trained with or without OR on CIFAR-100.

| Encoder | VIT-tiny | | VIT-small | | VIT-base | |
|---|---|---|---|---|---|---|
| Regularizer | Top-1 | Top-5 | Top-1 | Top-5 | Top-1 | Top-5 |
| - | 54.16 | 83.37 | 62.02 | 87.80 | 64.12 | 88.83 |
| SO | **60.84** | **87.65** | **64.95** | **89.47** | **66.91** | **90.42** |

To evaluate OR on more SSL methods, under the LightSSL framework, we test SO on CIFAR-10, IMAGENET-100, and IMAGENET-1k. OR still consistently improves the performance of various SSL methods as shown in Table 4.

Table 4: Performance of SSL methods on LightlySSL. ResNet18 is used on CIFRA-10, CIFAR-100 and IMAGENET-100, and ResNet50 is employed on IMAGENET-1K.

| Methods | CIFAR-10 (Epoch 200) | | CIFAR-10 (Epoch 400) | | IMAGENET-100 | | IMAGENET-1K | |
|---|---|---|---|---|---|---|---|---|
| | original | with SO | original | with SO | original | with SO | original | with SO |
| Barlow Twins | 83.58 | **84.78** | 85.91 | **86.25** | 56.6 | **57.0** | - | - |
| BYOL | 86.94 | **87.01** | 89.64 | **90.02** | 51.7 | **52.1** | - | - |
| DCL | 83.38 | **84.10** | 85.95 | **86.27** | - | - | - | - |
| DCLW | 82.42 | **82.73** | 85.25 | **85.71** | - | - | - | - |
| DINO | 81.87 | **81.95** | - | - | - | - | 59.77 | **60.40** |
| Moco | 85.19 | **85.32** | - | - | - | - | - | - |
| NNCLR | 82.31 | **82.34** | - | - | - | - | - | - |
| SimCLR | 84.55 | **84.84** | - | - | - | - | - | - |
| SimSiam | 79.27 | **84.31** | - | - | - | - | - | - |
| SwaV | 83.10 | **83.67** | - | - | - | - | - | - |

## 6.2 OR Works on Large-scale Dataset

We demonstrate the effects of OR on the large-scale dataset IMAGENET-1k. Specifically, we pre-train three BYOL models- BYOL without OR, BYOL with SO, and BYOL with SRIP on IMAGENET-1k

(with ResNet50). For each testing classification dataset, we report the accuracy as shown in Table 5 and 6.

Table 5: Classification and objection detection performance. BYOL is trained with or without OR on IMAGENET-1k (ResNet50 with batchsize 128, Epoch 100). The best results are in **bold**, the second best in *italics*.

| Dataset | IMAGENET-1k | |
|---|---|---|
| Regularizer | Top-1 | Top-5 |
| - | 65.81 | 87.06 |
| SO | *67.84* | *88.18* |
| SRIP | **67.91** | **88.20** |

| Dataset | VOC 2007+2012 | | |
|---|---|---|---|
| Pretraining methods | AP | AP50 | AP75 |
| Scratch | 33.8 | 60.8 | 33.1 |
| BYOL without OR | *44.74* | *76.04* | *46.24* |
| BYOL with SO | **53.81** | **81.46** | **59.43** |

Table 6: Classification accuracy on transfer learning datasets.

| Dataset | Food101 | | Flowers102 | | DTD | | GTSRB | | CIFAR-10 | | CIFAR-100 | |
|---|---|---|---|---|---|---|---|---|---|---|---|---|
| Regularizer | Top-1 | Top-5 | Top-1 | Top-5 | Top-1 | Top-5 | Top-1 | Top-5 | Top-1 | Top-5 | Top-1 | Top-5 |
| - | 59.838 | 83.945 | 73.817 | 91.039 | 68.777 | 91.596 | 63.539 | 92.035 | 79.7 | 99.12 | 52.09 | 81.89 |
| SO | *63.624* | **86.444** | *78.956* | *93.397* | **70.851** | **92.819** | *68.725* | *93.991* | *83.58* | **99.41** | *57.57* | *85.38* |
| SRIP | **63.628** | 86.420 | **79.33** | **94.243** | *70.426* | 92.234 | **69.256** | **94.347** | **84.26** | *99.39* | **57.84** | **85.87** |

We can observe that OR not only improves the accuracy on IMAGENET-1k but also on all the transfer learning datasets. OR improves TOP-1 accuracy by $3\%$ in IMAGENET-1k and by $3\%$ to $9\%$ in each transfer learning datasets. The transfer learning task evaluates the generality of the encoder as it has to encode samples from various out-of-distribution domains with categories that it may not have seen during pretraining. OR also significantly improves SSL's performance in the objection detection task by $20\%$ on AP. The above results are close to Chen et al. (2021), Da Costa et al. (2022), Weng et al. (2023) where also training only 100 epochs.

Table 7: Classification accuracy on IMAGENET-1k (pretrained with different epochs and batchsizes).

| Dataset | IMAGENET-1k | | | |
|---|---|---|---|---|
| Pretraining settings | Epoch 100 batchsize 128 | | Epoch 200 batchsize 256 | |
| Regularizer | Top-1 | Top-5 | Top-1 | Top-5 |
| - | 65.81 | 87.06 | 69.76 | 89.10 |
| SO | **67.84** | **88.18** | **70.16** | **89.47** |

Considering that training epoch and batchsize during pretraining significantly impact the performance Chen et al. (2021), Huang et al. (2024), Lavoie et al. (2022), we further increase the training epoch(200) and batchsize(256) and scale up the learning rate accordingly. As shown in Table 7, OR consistently improves the performance.

## 7 Conclusions

The existing studies focus on the dimensional collapse of representations and overlook whether weight matrices and hidden features also undergo dimensional collapse. We first time propose a mitigation approach to employing orthogonal regularization (OR) across the encoder, targeting both convolutional and linear layers during pretraining. OR promotes orthogonality within weight matrices, thus safeguarding against the dimensional collapse of weights, hidden features, and representations. Our empirical investigations demonstrate that OR significantly enhances SSL method performance across diverse benchmarks, yielding consistent gains with both CNNs and Transformer-based architectures as the backbones. Importantly, the time complexity and required efforts on fine-tuning are low and the performance improvement is significant, enabling it to become a useful plug-in in various SSL methods.

In terms of future research, we wish to examine the effect of OR on other pre-training foundation models, such as vision generative SSL models such as MAE (He et al. 2022), auto-regression models like GPTs and LLaMAs (Radford et al. 2018, 2019, Brown et al. 2020, Touvron et al. 2023), and Contrastive Language-Image Pre-training models (Radford et al. 2021, Li et al. 2022). We believe OR is a pluggable and useful module to boost the performance of vision and language foundation models. In fact, this paper is the first to test the effectiveness of OR in a Transformer-based architecture and it is reasonable to believe that it will perform well in these domains.

## Acknowledgments

The work described in this paper was supported by the National Natural Science Foundation of China (No. 52102385), grants from the Research Grants Council of the Hong Kong Special Administrative Region, China (Project No. PolyU/25209221 and PolyU/15206322), and grants from the Otto Poon Charitable Foundation Smart Cities Research Institute (SCRI) at the Hong Kong Polytechnic University (Project No. P0043552). The contents of this article reflect the views of the authors, who are responsible for the facts and accuracy of the information presented herein.

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

# A   Appendix / supplemental material

## A.1   Effects of Representation Whitening on the Encoder

In this section, we explore the effect of whitening representations on hidden features and weight matrices in the encoder. To be specific, similar to the settings of 5, we train three VICREG Bardes et al. (2022) models: original VICREG, VICREG without projector  (Li, Chen & Yang 2020), and VICREG with OR.

Original VICREG adds two regularization terms (variance and covariance regularization) to whiten the projector features. We use $X_{aug1}$ as an example to introduce them.

**Variance regularization.**  The variance regularization term ensures that each dimension of the learned representation $Z$ maintains a non-trivial variance. This is critical to prevent the collapse of dimensions, where a model might ignore certain informative variations in the data. Mathematically, the variance regularization can be expressed as follows:

$$L_{\text{var}} = \frac{1}{D} \sum_{d=1}^{D} \max(0, \gamma - \text{S}(z_d, \epsilon)) \tag{5}$$

where $D$ is the dimensionality of $Z_{aug1} = f(X_{aug1})$, $z_d$ represents the $d$-th dimension of $Z_{aug1}$, $\text{S}(z_d, \epsilon) = \sqrt{\text{Var}(z_d)) + \epsilon}$ is the regularized standard deviation of $z_d$ across different samples, and $\gamma$ is a threshold parameter that dictates the minimum desired standard deviation for each dimension.

**Covariance regularization.**  The covariance regularization term is designed to decorrelate the different dimensions of $Z_{aug1}$. By minimizing the off-diagonal elements of the covariance matrix of $Z_{aug1}$, this term helps ensure that different dimensions capture distinct aspects of the data, thereby preventing redundancy in the representation. The covariance regularization is defined as:

$$L_{\text{cov}} = \sum_{i \neq j} \left( \text{Cov}(z_i, z_j) \right)^2 \tag{6}$$

where $\text{Cov}(z_i, z_j)$ denotes the covariance between the $i$-th and $j$-th dimensions of $Z_{aug1}$. This term effectively encourages the representation to have orthogonal dimensions, which is beneficial for learning independent features.

As for the VICREG without projector, we discard the projector and apply the SSL objective directly to the representations, which ensures that the representations are whitened (no dimensional collapse in representations), i.e., minimize the correlation among dimensions and make each dimension rich in information. For OR, we choose SO as the regularizer and set $\gamma$ as $1e - 6$. We then experimentally observe that guaranteeing that dimensional collapses do not occur in representations or projector features (i.e., VICREG without projector and projector features) does not guarantee that dimensional collapses do not occur in weight matrices in the encoder. Moreover, discarding the projector even damages the performance of the original VICREG, while OR still boosts the performance as shown in Table  8.

Table 8: Performance comparison of different VICREG configurations.

| Models | VICREG without Projector | VICREG | VICREG with SO |
|--------|--------------------------|--------|----------------|
| Top-1  | 88.64 | 91.64 | **92.35** |
| Top-10 | 99.57 | 99.73 | **99.79** |

As shown in Figure 4, the weight matrices of the original VICREG suffer from dimensional collapse and whitening representations directly (i.e., getting rid of the projector) even makes the eigenvalue decay faster for the weight matrices. OR can alleviate this phenomenon.

## A.2   Visualization of Weight Matrices

In this section, layer4 of ResNet18 pretrained with BYOL on CIFAR-10 is visualized. To be specific, we calculate the correlation coefficient matrix of the weight matrix and then plot the HeatMap of the correlation coefficient matrix and the results of Spectral Biclustering. As shown in Figure  5,

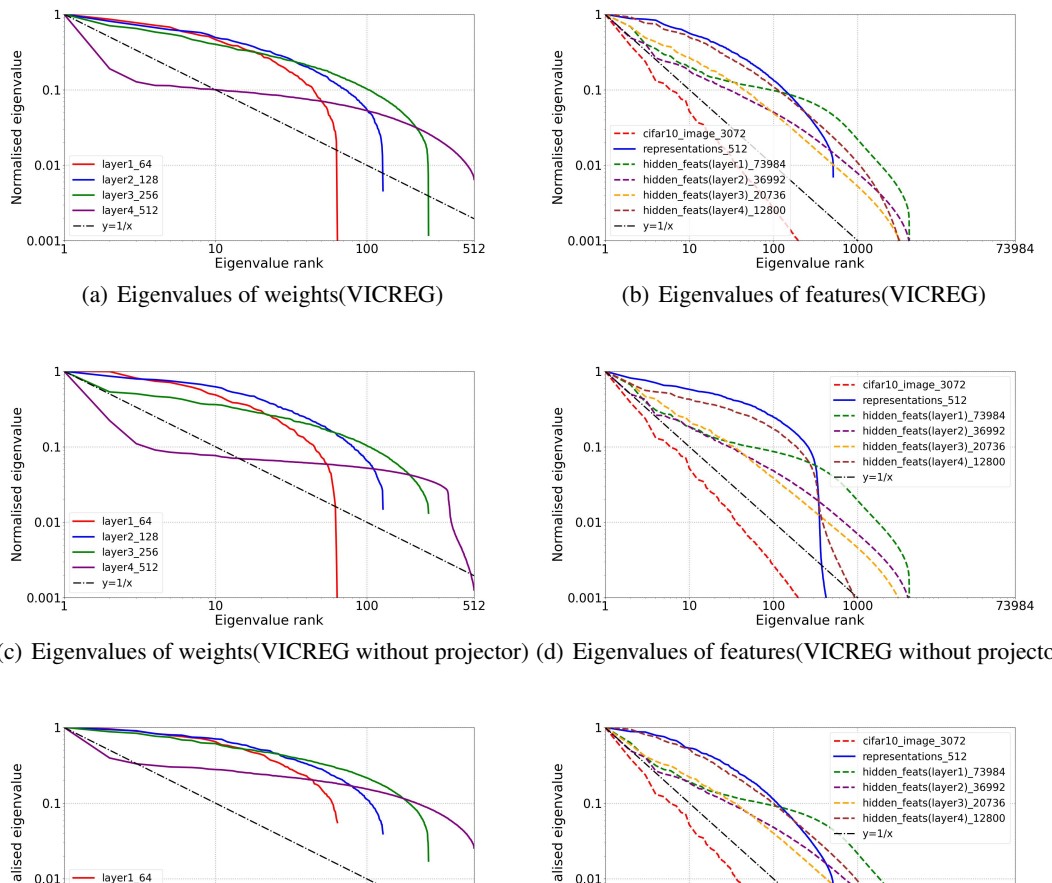

(a) Eigenvalues of weights(VICREG)  (b) Eigenvalues of features(VICREG)

(c) Eigenvalues of weights(VICREG without projector) (d) Eigenvalues of features(VICREG without projector)

(e) Eigenvalues of weights(VICREG with SO)  (f) Eigenvalues of features(VICREG with SO)

Figure 4: Eigenspectra of both weights and features within the encoder (ResNet18). The features are collected on the first batch of the test set (batchsize 4096). We pretrain original VICREG, VICREG without projecto, and VICREG with OR on CIFAR-10. The x-axis and y-axis are both log-scaled. The solid line represents that all eigenvalues are positive, the dashed line represents the existence of eigenvalues that are non-positive, and the number of eigenvalues is represented behind the underline.

HeatMap can intuitively indicate that the correlation of non-diagonal elements is constrained to 0 by OR. The Biclustering results show an obvious blocky structure, which means that there is clustering between filters, and the weight matrix is low-rank and redundant.

### A.3  Visualization of Representations

We used BYOL (ResNet18) for pretraining on CIFAR-10. After pretraining, we perform dimension reduction and visualization of learned representations using UMAP (McInnes et al. 2018). As shown in Figure 6, in the absence of OR, there is a tendency for the cluster centers of each category to move closer together and more outliers appear. This is due to the fact that in the absence of OR, BYOL produces representations dominated by some extremely large eigenvalues (i.e. dimensional collapse), which is consistent with results in Section 5.2.

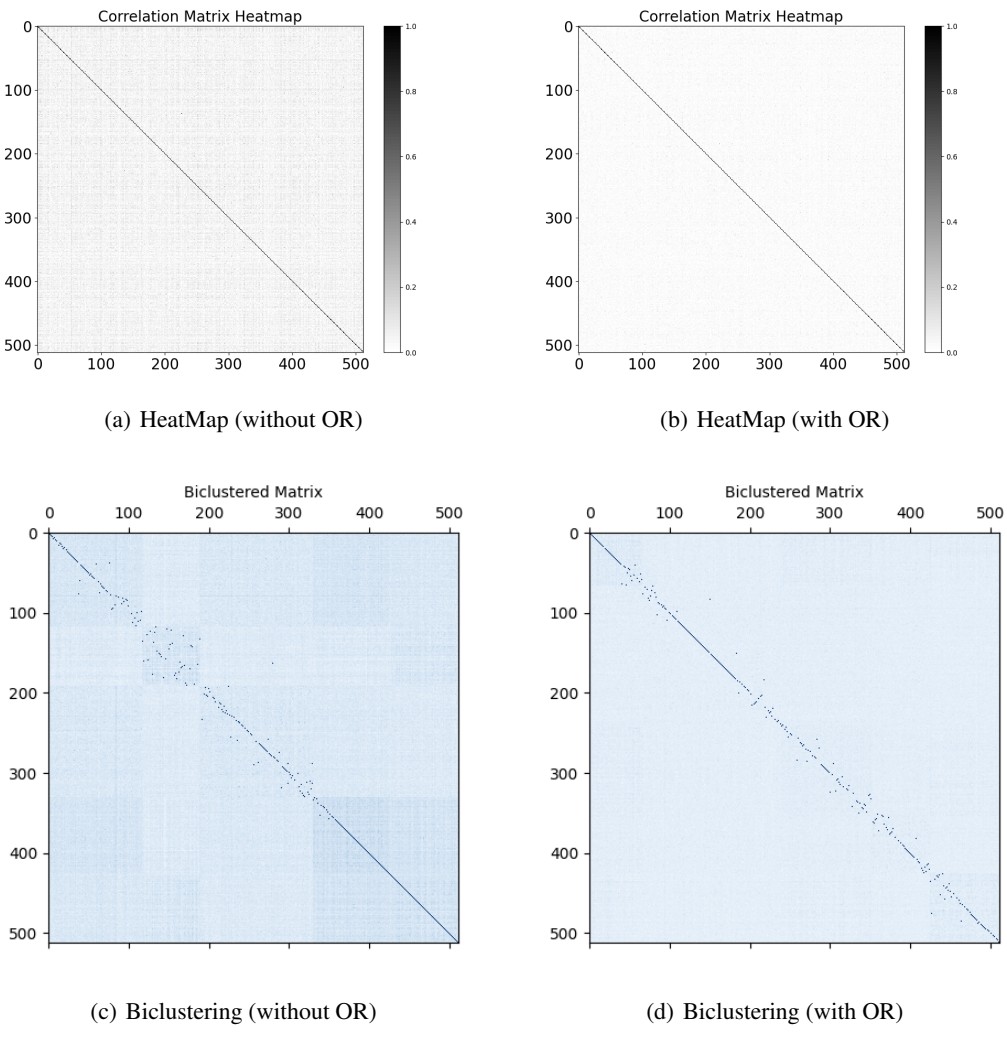

(a) HeatMap (without OR)

(b) HeatMap (with OR)

(c) Biclustering (without OR)

(d) Biclustering (with OR)

Figure 5: HeatMap is the visualization of the absolute value of the correlation coefficients among filters of the weight matrix (layer4). Biclustering is the visualization of the results of spectral biclustering. It can be seen that OR significantly reduces the correlation and removes the clustering patterns among filters from the heatmap and biclustering, respectively.

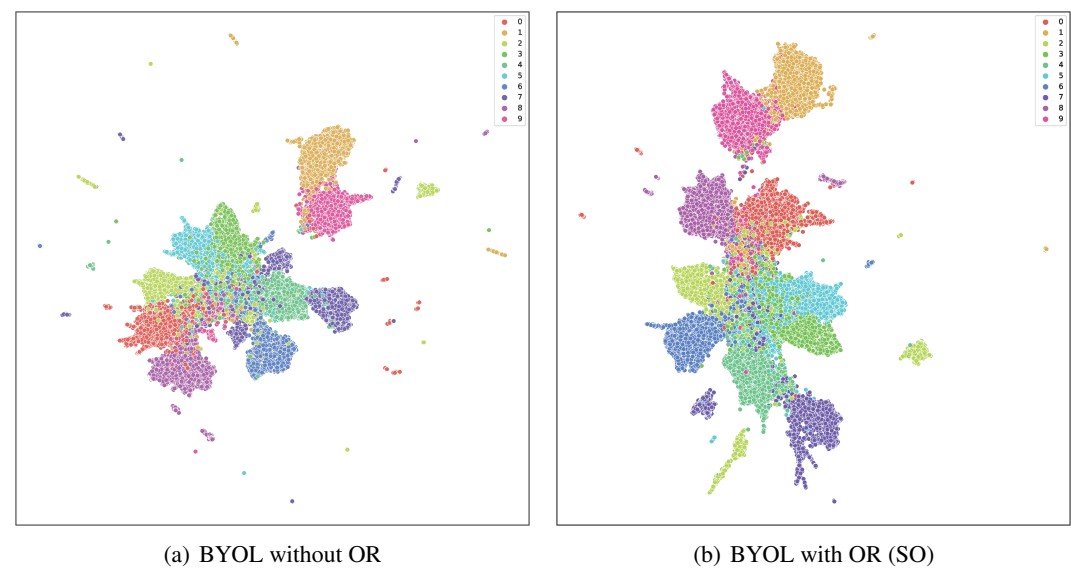

(a) BYOL without OR

(b) BYOL with OR (SO)

Figure 6: Visualization of Representations

### A.4 Datasets

We utilized several datasets for pretraining and evaluating SSL methods. Below we provide a detailed description of these datasets:

- **IMAGENET-1k** (Deng et al. 2009): A large dataset contains 1,281,167 training images, 50,000 validation images, and 100,000 test images, which spans 1000 object classes.

- **IMAGENET-100** (Deng et al. 2009): A subdataset of IMAGENET-1K, containing 100 classes with 1000 training data and 300 test data per class.

- **CIFAR-10** (Krizhevsky et al. 2009): Comprising 60,000 images in 10 classes, with each class containing 6,000 images. The split includes 50,000 training images and 10,000 test images.

- **CIFAR-100** (Krizhevsky et al. 2009): This dataset consists of 60,000 images divided into 100 classes, with 600 images per class. The dataset is split into 50,000 training images and 10,000 test images.

- **Food-101** (Bossard et al. 2014): This dataset includes 101,000 images of food dishes categorized into 101 classes, with each class having approximately 1,000 images.

- **Flowers-102** (Xia et al. 2017): Contains 8,189 images of flowers from 102 different categories. Each class consists of between 40 and 258 images.

- **DTD** (Sharan et al. 2014): The Describable Textures Dataset (DTD) includes 5,640 images categorized into 47 different texture categories.

- **GTSRB** (Haloi 2015): The German Traffic Sign Recognition Benchmark (GTSRB) dataset consists of over 50,000 images of traffic signs across 43 categories.

- **PASCAL VOC2007 and VOC2012** (Lin et al. 2014): Used for evaluating objection tasks, this dataset includes complex everyday scenes with annotated objects in their natural context. The objection detection task contains 20 categories. We use the VOC2007 and VOC2012 train-val (16551 images) as the training set and then report the performance on the VOC2007 test set (4952 images).

Each dataset was carefully curated to support the training and validation of our models, ensuring a comprehensive evaluation across various image classification and segmentation tasks.

### A.5 Joint-embedding SSL methods

Self-supervised learning (SSL) has emerged as a powerful paradigm for learning representations without the need for labeled data. This appendix provides a concise overview of several SSL methods used in this paper.

- MOCOv2, introduced by Chen & He (2021), on top of MOCO's momentum encoder and the use of the dynamic dictionary with a queue to store negative samples (He et al. 2020), adds the MLP projection head and more data augmentation. Compared to MOCOv2, MOCOv2plus uses a symmetric similarity loss.

- MoCov3 (Chen et al. 2021) makes some improvements on the basis of v1/2, firstly, because the batchsize is large enough when training V3, the memory queue is removed, and the negative samples are sampled directly from the batch. Secondly, symmetric contrastive loss is used, and finally, an extra prediction head is added to the original encoder, which is a two-layer fully connected layer.

- Bootstrap Your Own Latent (BYOL), proposed by Grill et al. (2020), introduces a novel approach to SSL that does not rely on negative pairs. Instead, BYOL employs a dual-network architecture where the encoder learns to predict the representations of the momentum encoder. Through a series of updates (i.e. EMA), where the momentum encoder gradually assimilates the encoder's weights, BYOL effectively learns robust representations. The success of BYOL depends not only on the EMA, but also on its additional projector and the BN in the projector to avoid a complete collapse of the encoder. This method challenges the conventional wisdom that contrastive learning requires negative pairs, opening new avenues for SSL research.

- Expanding on the ideas of BYOL, NNBYOL (Dwibedi et al. 2021) introduces the concept of using nearest neighbors to augment the learning process. By leveraging the similarities between different instances in the dataset, NNBYOL aims to refine the quality of the learned representations further. This approach underscores the potential of incorporating instance-level information into the SSL framework, enhancing the discriminability and robustness of the resulting models.

- DINO (Caron et al. 2021) uses a self-distilling architecture. The outputs of the teacher networks (i.e. the momentum encoder) are subjected to a centering operation by averaging over a batch, and each network outputs a K-dimensional feature that is normalized using Softmax. The similarity between the student model (i.e. the encoder) and the teacher model is then computed using cross-entropy loss as the objective function. A stop-gradient operator is used on the teacher model to block the propagation of the gradient, and only the gradient is passed to the student model to make it update its parameters. The teacher model is updated using the weights of the student model (i.e. EMA).

- Barlow Twins computes the correlation matrix between the embeddings of two different views of the same sample and avoids collapse by making it as close as possible to the unit matrix. This approach makes the embeddings between the two views of the sample as similar as possible while minimizing the redundancy between vector components.

- VICREG avoids the complete collapse problem with variance and covariance regularization.

- DCL and DCLW removes the NPC effect of infoNCE loss by getting rid of the positive term from the denominator and thus significantly improves the learning efficiency

- SimSiam achieves a very strong baseline without using large batchsize, negative samples, or momentum encoder using only the stop-gradient operation.

- SwaV is trained by predicting the clustering assignment of another view and also introduces multi-crop, which increases the number of views by reducing the image size without increasing the extra memory and computational requirements.

- SimCLR establishes a simple and effective architecture for contrastive learning by increasing the batchsize, augmenting the data and adding a nonlinear projector after the representation.

## A.6 Hyper-parameters of Pretraining and Evaluation

For each SSL method, we use the original settings of Solo-learn and LightlySSL (Da Costa et al. 2022). These settings include the network structure, loss function, training policy, and data augmentation policy. Considering that we use numerous SSL methods and that our setup is exactly the same as them, please go to their official implementation.

For OR, the appropriate regularization term $\gamma$ generally depends only on the backbone used by SSL and the orthogonality regularizer (SRIP or SO) chosen. As shown in Table 9, when you want to add OR to your SSL pre-training, you simply pass the encoder into the loss function, and then you just need to set $\gamma$ of the OR according to the backbone and regularizer you use.

Table 9: The recipe of adding OR

| Encoder | Regularizer | Regularization term |
|---------|-------------|---------------------|
| ResNet18 | SO | 1e-6 |
|  | SRIP | 1e-3 |
| ResNet50 | SO | 1e-6 |
|  | SRIP | 1e-3 |
| WideResnet28w2 | SO | 1e-6,1e-7 |
|  | SRIP | 1e-4,1e-5 |
| VIT-tiny | SO | 1e-5 |
| VIT-small | SO | 1e-5 |
| VIT-Base | SO | 1e-6 |

For the classification tasks, due to computational constraints, we do not perform non-linear fine-tuning in classification tasks. Instead, we perform a linear probe or KNN to evaluate the quality of obtained representations as typically done in the literature (Huang et al. 2024, Li, Chen & Yang 2020, Lavoie

et al. 2022). To be specific, for each SSL method and dataset, after pretraining, We train a linear classifier on top of frozen representations of the training set. Then we report the Top-1 and Top-5 linear classification accuracy on the test set. When training the linear classifier, we use 100 epochs, weight decay to 0.0005, learning rate 0.1 (we divide the learning rate by a factor of 10 on Epoch 60 and 100), batchsize 256, and SGD with Nesterov momentum as optimizer (In IMAGENET-1k, we use batchsize 128 and learning rate 0.2).

For the object detection task, we perform nonlinear fine-tuning on ResNet50 in RCNN-C4 (Girshick et al. 2014) with batchsize 9 and base learning rate 0.01. We use the detectron2 (Wu et al. 2019), following the MOCO-v1 (He et al. 2020) official implementation exactly.

## A.7   Time Cost of OR

Implementing OR requires computing the OR loss in the backbone at each gradient update, we count the time overhead required by the different backbones to compute OR at one time, and we have averaged over 10 times as shown in Table A.7. In the pre-training phase, the time overhead of OR is only related to the backbone and the steps that need to be updated, IMAGENET-1k (100 epochs, batchsize 128) has a total of 62599 steps, and CIFAR-100 (1000 epochs, batchsize 256) has a total of 194999 steps. As you can see, compared to the original pre-training overhead of dozens and hundreds of hours, the additional time added by OR is very small, steadily improving SSL's performance. Notably, if we use a larger batchsize such as 4096, our time overhead will be reduced by 64 on IMAGENET-1k and 16 on CIFAR-100.

Table 10: Time cost of OR

| Encoder | | ResNet18 | ResNet50 | WideResNet28w2 | VIT-tiny | VIT-small | VIT-base |
|---|---|---|---|---|---|---|---|
| Single step | Overhead of SO | 0.016s | 0.022s | 0.008s | 0.019s | 0.024s | 0.045s |
| | Overhead of SRIP | 0.012s | 0.030s | 0.012s | 0.029s | 0.034s | 0.054s |
| CIFAR-100 | Original(DINO) | 12h 26m | 18h 10m | 7h 36m | 8h 11m | 13h 18m | 1d 8h |
| | Overhead of SO | 0.87h | 1.19h | 0.43h | 1.03h | 1.30h | 2.44h |
| | Overhead of SRIP | 0.64h | 1.62h | 0.65h | 1.57h | 1.84h | 2.92h |
| IMAGENET-1k | Original(BYOL) | - | 3d 17h | - | - | - | - |
| | Overhead of SO | - | 0.38h | - | - | - | - |
| | Overhead of SRIP | - | 0.52h | - | - | - | - |

