# OpenReview forum: "Preventing Dimensional Collapse in Self-Supervised Learning via Orthogonality Regularization"
_NeurIPS.cc/2024/Conference — NeurIPS 2024 poster_

### Official Review · Reviewer_Yvo9 · 2024-07-01

**Soundness:** 2
**Presentation:** 2
**Contribution:** 1
**Rating:** 5
**Confidence:** 4

**Summary:**

Driven by the dimensional collapse phenomenon observed in self-supervised learning, this research adopts a methodology designed to prevent the reduction in dimensionality of weight matrices. The empirical findings indicate a reduction in dimensional collapse in certain instances.

**Strengths:**

The proposed method seamlessly integrates with existing SSL methods.

**Weaknesses:**

The proposed method lacks novelty, essentially applying the loss formulation from Barlow Twins to the weight matrix.

Moreover, the experimental results are insufficient to substantiate the efficacy of the method due to the following reasons:

1. Many of the SSL methodologies utilized to assess the OR are no longer considered state-of-the-art. Therefore, it is recommended to evaluate the OR using state-of-the-arts methods.

2. Table 1 reveals that the effectiveness of SO or SRIP appears to exhibit considerable variability, rendering the proposed methods unstable.

3. Tables 3 and 5 exclusively demonstrate the evaluation of OR for the BYOL method, omitting the efficacy assessments of other SSL methods. Additionally, in Table 4, the datasets are characterized as small-scale, which contradicts the claims made by the authors.

**Questions:**

1. What are the advantages of regularizing weights over regularization of representation?

2. What is the performance of SRIP in Table 2 and 5?

**Limitations:**

Please refer to the weakness.

---

> ### Author Rebuttal · Authors · 2024-08-06
>
> We appreciate your comments and feedback. In addition to the general response, we address your itemized concerns here.
>
> >The proposed method lacks novelty, essentially applying the loss formulation from Barlow Twins to the weight matrix.
>
> Methods like Barlow Twins try to avoid dimensional collapse of intermediate features as well as weights by whitening the projector output.
> Our OR, on the other hand, acts directly on the weight matrix to indirectly avoid the dimensional collapse of features.
> The implementation and motivation of these two methods are different, and no one has previously explored the dimensional collapse of the weight matrix in SSL, nor has anyone attempted to constrain the weight matrix in SSL to obtain an improvement in the results (except for weight decay).
>
> In a broader sense, OR is just one of the methods we use to constrain the weight matrix in SSL, as it is intuitive to avoid the collapse of the weight matrix and has deterministic effects on the features. We only employed the widely used form of OR, but we wanted to share it with the community given its plug-and-play and interpretability in SSL methods.
>
> We complement the experiments based on feature whitening methods such as Barlow Twins and VICRGE, OR can further improve their performance. We also visualize and analyze BYOL w/ vic, VICREG, and VICREG w/ OR, which are not able to avoid dimensional collapse of their weight matrices by feature whitening alone. Simply adding feature whitening to the original SSL method will even degrade its performance, unlike our plug-and-play OR.
> Please check the performance table and see the pdf material of the general response.
>
> >Many of the SSL methodologies utilized to assess the OR are no longer considered state-of-the-art. Therefore, it is recommended to evaluate the OR using state-of-the-arts methods.
>
> Thank you for your advice. The SSL methods we have now include **6** methods implemented by solo-learn (MOCOv2plus, MOCOv3, DINO, NNBYOL, BYOL, VICREG) and **10** methods implemented by the LIGHTLY framework (BarlowTwins, BYOL, DCL, DCLW, DINO,  Moco, NNCLR, SimCLR, SimSiam, SwaV). Removing duplicates, there are in total **13** different SSL methods.
> Please check the performance table in the general response.
>
> We are currently experimenting as many SSL methods as possible with that have been built as benchmarks for fair comparison (LIGHTLY, solo-learn). If there are  SOTA SSL methods that we still need to include, please let us know and we will try to add OR to it.
>
> >Table 1 reveals that the effectiveness of SO or SRIP appears to exhibit considerable variability, rendering the proposed methods unstable.
>
>
> Our ORs, both SO and SRIP, work on all kinds of SSLs and all kinds of backbones. Only when MOCOv2 plus replaces the backbone with ResNet50, OR makes the Top-1 go down, but its Top-5 all goes up.
> Please note that we did not adjust any hyperparameters when changing the backbone or adding OR.
>
> >Tables 3 and 5 exclusively demonstrate the evaluation of OR for the BYOL method, omitting the efficacy assessments of other SSL methods.
>
> This is because training on large-scale data takes too much time (3 days for 100 epochs), and then BYOL presents the most stable and best performance on cifar100, so we only tested BYOL for now.
> Due to our limited time and computational resources, the results of DINO and DINO (SO) on ImageNet with larger batchsize (2048) and epoch (300) are provided.
> Also, we report  BarlowTwins and BYOL results under Imagenet100 (ResNet 18, epoch 200, batch size 256).
> Please check the performance table in the general response.
>
>
> >Additionally, in Table 4, the datasets are characterized as small-scale, which  contradicts the claims made by the authors.
>
> Table 4 refers to the results of SSL models pre-trained on large-scale datasets to perform linear probing on small datasets, which is consistent with numerous SSL studies. We will add more descriptions to the table later to avoid ambiguity.
>
>
>
> >What are the advantages of regularizing weights over regularization of representation?
>
> We supplement the experiments based on feature whitening methods such as Barlow Twins and VICRGE, **OR can further improve their performance**. We also visualize and analyze BYOL w/ vic, VICREG, and VICREG w/ OR, which **are not able to avoid dimensional collapse of their weight matrices by feature whitening alone**. Simply adding feature whitening to the original SSL method will **even degrade** its performance, unlike our plug-and-play OR.
> Please check the performance table and see the pdf material of the general response.
>
> >What is the performance of SRIP in Table 2 and 5?
>
> The VITs experiment in Table 2 and the BYOLl 200 epoch experiment in Table 5 were too time-consuming, so we only tested the effect of SO. The results of SO are very similar to those of SRIP, with shorter time and less memory usage. In general, we advocate the use of SO for large models. However, we will still add the results of SRIP with the same settings in the later version.

---

> > ### Comment · Reviewer_Yvo9 · 2024-08-10
> >
> > Thank you for conducting the additional experiments and providing the discussions in response to the review. The authors' rebuttal has addressed most of the previously raised concerns, leading me to lean towards a positive review score. If the authors emphasize their responses regarding the novelty of the paper and the advantages of regularizing weights over regularizing representations in the introduction, it will likely enhance the paper's novelty and motivation. Therefore, I hope these responses will be incorporated into the manuscript.

---

> > > ### Author Response · Authors · 2024-08-10
> > >
> > > Thank you so much for recognizing our work! Because of your detailed and kind suggestions, we have greatly improved the quality and novelty of our articles, and we will definitely add these to the manuscript.

---

### Official Review · Reviewer_2nFS · 2024-07-12

**Soundness:** 3
**Presentation:** 3
**Contribution:** 2
**Rating:** 6
**Confidence:** 4

**Summary:**

In order to combat the problem of dimensionality collapse in self-supervised learning, the authors of this work introduce orthogonal regularization on encoders during pretraining. Taking inspiration from previous work on supervised orthogonality regularization, the authors study the impact of OR on eigenvalues of weights when considering self-supervised learning and find that OR learned weights/features contain more meaningful features and avoid dimensional collapse better. The recommended result seems to be effective for both contrastive and non-contrastive SSL methods.

**Strengths:**

Overall I think the idea is simple and the paper is well written. Particularly the observations regarding the eigenvalues of SSL features and weights and the impact of OR are informative. In addition, I believe the OR improvements on transfer learning and downstream tasks are considerable and could become a staple part of SSL.

**Weaknesses:**

1) I would like to point out the improvements of OR on SSL methods is not as major on the simple datasets such as Cifar100 set according to Table 1. However, the changes in larger datasets with BYOL are more noticeable but I would like to ask why results for other methods were not included for larger datasets.
2)  Particularly with regards to Fig 3, to me it doesn’t look as if the feature eigenvalue changes much with the OR except for the representations. Perhaps a combination of other OR methods could help ? Or adding orthogonality regularization with the features.
3) With regards to classification accuracy on Table 5, higher batch size and longer training somewhat diminish the impact of OR improvement. Perhaps this result could be used to encourage OR when training with smaller batches or less epochs is required ?

**Questions:**

1) For section 3.2 under what case would we have input > output ?
2) Could the authors provide some explanation for what entails whitening of the features ?

---

> ### Author Rebuttal · Authors · 2024-08-06
>
> We appreciate your comments and feedback. In addition to the general response, we address your itemized concerns here.
>
> >I would like to point out the improvements of OR on SSL methods is not as major on the simple datasets such as Cifar100 set according to Table 1\. However, the changes in larger datasets with BYOL are more noticeable but I would like to ask why results for other methods were not included for larger datasets.
>
> This is because training on large-scale data takes too much time (3 days for 100 epochs), and then BYOL presents the most stable and best performance on cifar100, so we only tested BYOL for now.
> Due to our limited time and computational resources, the results of DINO and DINO (SO) on ImageNet with larger batchsize (2048) and epoch (300) are provided.
> Also, we report  BarlowTwins and BYOL results under Imagenet100 (ResNet 18, epoch 200, batch size 256).
> Please check the performance table in the general response.
>
> >Particularly with regards to Fig 3, to me it doesn’t look as if the feature eigenvalue changes much with the OR except for the representations. Perhaps a combination of other OR methods could help ? Or adding orthogonality regularization with the features.
>
> Interestingly, the effects of OR on intermediate features do not appear to be as large as that of representations.
> We visualize intermediate features of VICREG (based on feature whitening), and VICREG\_OR. Although OR can improve the performance of VICREG, we can find that the change in their intermediate features and even the representation is not obvious, but the weight matrix has a significant change.
> Perhaps we need to train a deeper network to observe the effect of OR on the deeper features, we leave this part for future work.
> Please check the performance table and see the pdf material of the general response.
>
> >With regards to classification accuracy on Table 5, higher batch size and longer training somewhat diminish the impact of OR improvement. Perhaps this result could be used to encourage OR when training with smaller batches or less epochs is required ?
>
> Thank you for your suggestions\! Our current results on Imagenet show that SO is able to improve more with small batchsize and small epochs.
> To test the effects of OR with large batchsize and long epochs, the results of DINO and DINO (SO) on ImageNet with larger batchsize (2048) and epoch (300) are provided.
> In the follow-up work, we will definitely make up more epoch experiments to fully test the effect of OR.
> In addition, we added the results of various SSL methods on LIGHTLY framework cifar10, (resnet18), 200 epochs, and 400 epochs, the OR can get the same improvement.
> Please check the performance table in the general response.
>
> >For section 3.2 under what case would we have input \> output ?
>
> For convolutional layers, input \= convolution kernel area \* input feature dimension, output \= output feature dimension, e.g. for the first convolution in ResNet18, input \= 7\*7\*3, output \= 64, and input\>output.
> For linear layers, input is greater than output as long as the input feature dimension is greater than the output feature dimension.
>
> >Could the authors provide some explanation for what entails whitening of the features?
>
> OR causes features to whiten because the orthogonalized weight matrices, which make the feature spectra more consistent across depth layers of features, prevent deeper features (e.g., representations) from collapsing and showing a whitened nature \[1,2\].
> We can observe that the distribution of features in each layer is much closer after adding the OR constraints (Table 3 (d) brown and yellow lines in the paper).
>
> \[1\] Huang, Lei, et al. "Orthogonal weight normalization: Solution to optimization over multiple dependent stiefel manifolds in deep neural networks." *Proceedings of the AAAI Conference on Artificial Intelligence*. Vol. 32\. No. 1\. 2018\.
> \[2\] Wang, Jiayun, et al. "Orthogonal convolutional neural networks." *Proceedings of the IEEE/CVF conference on computer vision and pattern recognition*. 2020\.

---

### Official Review · Reviewer_sksr · 2024-07-12

**Soundness:** 3
**Presentation:** 2
**Contribution:** 2
**Rating:** 5
**Confidence:** 4

**Summary:**

This work proposes a method to reduce the dimensional collapse issue in self-supervised pretrained networks. Dimensional collapse occurs when a few large eigenvalues dominate the eigenspace. While previous research focused solely on the output representations of the pretrained networks, this work shows that dimensional collapse also occurs on the intermediate representations and weight matrices.

To address this issue, the authors use orthogonal regularization (OR) objectives for the weight matrices in linear layers (dense or convolutional) within the network. The OR objective promotes orthogonality in the weight matrices by minimizing the distance between the Gram matrix of each weight matrix and the identity matrix. They demonstrate that using the OR regularization objective leads to a smoother decay of eigenvalues in the (intermediate) representations and weight matrices of a self-supervised pretrained ResNet18.

The authors evaluate the OR regularization objective by pretraining various backbones (RN18, RN50, WideResNet28w2, and ViT-tiny/small/base) with multiple joint-embedding SSL methods (MoCov2, MoCov3, DINO, BYOL, NNBYOL) on CIFAR-100. They then train linear classification models on the frozen features, observing improved accuracy in nearly all cases with OR. Finally, they pretrain a RN50 backbone with BYOL on ImageNet-1k and evaluate it through linear probing on ImageNet and fine-tuning on Pascal object detection, showing improved results with OR.

**Strengths:**

- The used method of reducing dimensional collapse of SSL methods is simple and can be easily integrated on various SSL methods and backbones.
- The strongest aspect of this work is that it shows that OR consistently improves various joint-embedding SSL methods and backbones. However, the improvements (in linear accuracy) are small in the case of well-tuned baselines.

**Weaknesses:**

- **(W1)** The introduction's claimed contribution, "We **systematically study** the phenomenon of dimensional collapse in SSL, including how whitening representations and network depth affect the dimensional collapse of weight matrices and hidden features," seems to me to be an exaggeration. In the main paper, there is only one experiment pretraining ResNet18 with BYOL w/ and w/o OR on CIFAR-10, and another experiment in the supplementary material using VICReg, which are not directly comparable. Given the emphasis on this contribution, a more appropriate approach would be to include an analysis of whitening's impact on dimensional collapse in the main paper (e.g., adding another column with whitening-based results in Figure 3). This analysis should be conducted under the same conditions as the experiments in Figure 3, for example, using BYOL with a projector and whitening. This setup would enable a straightforward comparison of vanilla BYOL, BYOL with OR, and BYOL with whitening in terms of their effects on the eigenvalues of intermediate representations and weight matrices.
- **(W2)**: **Limited technical contribution**: as discussed in the related work sections of this work, orthogonality regularization of the weight matrices within neural network has been extensively studied before (related work subsection 2.3). In particular, the OR objectives studied here are directly taken from Bansal et al. 2018, and straight-forwardly applied to the examined self-supervised approaches, without any modification at all. So, the main novelty of this work is to apply in a plug-and-play way the OR regularization objectives of Bansal et al. 2018 on self-supervised methods.
- **(W3)** About the CIFAR100 results: in the results provided in solo-learn (https://github.com/vturrisi/solo-learn), which the authors also use for their experiments, some baselines (i.e., SSL methods that do not use OR) have higher results than what is reported in this work and often higher even to the results reported in this work with the OR regularization objective.  In particular, this is the case for MoCov2+ (69.89% in solo-learn vs 69.51% here), MoCov3 (68.83% in solo-learn vs 67.13% here), DINO (66.76% in solo-learn vs 59.13% in here). For BYOL however, is the other way around (70.46% in solo-learn vs 71.15% in here). What is the reason for the differences in the results?
- **(W4)** The improvement that OR offers over a well-tuned BYOL baseline on ImageNet (200 pretraining epochs and 256 batch-size) is marginal: from 69.76 to 70.16. It seems that it offers significant performance improvements in mostly un-tuned baselines, like the results when pretraining BYOL for 100 and for 64 batch-size. In reality, even 200 epochs are not adequate for pretraining BYOL that, its performance reaches 74.3 for 1000 epochs (see BYOL paper). So, it is unclear whether OR could offer any improvement for such long pretraining settings that are typically used in SSL.
- **(W5)** Not my primary concern, but although OR is shown to provide a small improvement on the linear accuracy of some SSL methods (e.g., going from 69.76 to 70.16 for BYOL pretrained with 200 epochs on ImageNet-1k), no evidence is shown that OR actually avoids dimension collapse in settings where otherwise (i.e., without OR) the SSL method will completely collapse and fail to learn any meaningful representation.

**Questions:**

I would like the authors to address my concerns (W1), (W2), (W3) and (W4), as described in the weaknesses section. Some more specific questions:
- About W1: can the authors provide a comparison of vanilla BYOL, BYOL with OR, and BYOL with whitening in terms of their effects on the eigenvalues of intermediate representations and weight matrices.
- About W3: What is the reason for the differences in the results reported in this work and in in solo-learn (https://github.com/vturrisi/solo-learn)?
- About W4: Can the authors demonstrate that OR improves even well-tuned baselines like BYOL with 1000 epochs (74.3 in BYOL paper) or MoCo-v2 with 800 epochs (71.1% in https://arxiv.org/pdf/2003.04297), or MoCo-v3 (74.6 for RN50 see: https://github.com/facebookresearch/moco-v3). Any of them would be good.

**Limitations:**

Yes, they are discussed.

---

> ### Author Rebuttal · Authors · 2024-08-06
>
> We appreciate your comments and feedback. In addition to the general response, we address your itemized concerns here.
>
> >(W1)
>
> Thank you for your very constructive suggestions\! Following your suggestions, we added more experiments of BYOL with vic loss (feature whitening their predictor's output). Simply adding feature whitening to the original SSL method will even degrade its performance, in contrast, our plug-and-play OR method can improve the performance.
> We also visualize weights and features of BYOL w/ vic, VICREG, and VICREG w/ OR (please see the pdf material), which are not able to avoid dimensional collapse of their weight matrices by feature whitening alone.
> Please check the performance table and see the pdf material of the general response.
>
> >(W2)
>
> Thanks for the comment, in which we respectfully disagree. Although OR has been explored under supervised and semi-supervised learning, this paper is the first time to explore its effect on dimensional collapse in SSL and attempt to turn it into a plug-and-play method for enhancing SSL. It is also the first time that OR has been applied to a transformer architecture.
> We only employed the widely used form of OR, but we wanted to share it with the community given its plug-and-play and interpretability in SSL methods.
>
> To further highlight our contribution in a more systematic study of dimensional collapse in SSL, we have added relevant experiments and visualizations based on your suggestions:
> BYOL, BYOL with OR, and BYOL with whitening in terms of their effects on the eigenvalues of intermediate representations and weight matrices.
>
> The contribution of this paper is rephrased and summarized as follows: From our observations, although existing methods such as VICREG deal with the collapse of features via whitening features, they can not prevent the dimensional collapse of the weight matrices. We are not only the first work to systematically study dimensional collapse at all levels in SSL but also provide new solutions to address the dimensional collapse by constraining the weight matrix.
>
> OR is just one of the methods we use to constrain the weight matrix in SSL, as it is intuitive to avoid the collapse of the weight matrix and has deterministic effects on the features.
> We only employed the widely used form of OR, but we wanted to share it with the community given its plug-and-play and interpretability in SSL methods.
>
> >(W3)
>
> Thanks for pointing this out. We followed the latest Solo-learn config file exactly and ran the results directly. We carefully compared the config in its GitHub repository with the config provided by checkpoints and found that there are differences in its hyperparameters, which is most likely the reason why our reproduced results are not the same as its report results.
>
> For example, in DINO, its default config uses the lars optimizer, warmup\_teacher\_temperature\_epochs: 0, warmup\_start\_lr: 3e-05. However, the optimizer used for reporting results is SGD, lars=Ture, warmup\_teacher\_temperature\_epochs: 50, warmup\_start\_lr: 3e-03.  These differences in hyperparameters result in our results being different from the results of their checkpoint reporting.
>
> Our baseline and OR methods run entirely under their officially provided hyperparameter configs. **It is difficult to make direct use of the parameters in its checkpoints. solo-learn has been updated and its hyperparameter form has been changed.**
>
> Fortunately, our OR methods are plug-and-play enough to compare the impact of SO with 10 SSL methods in another open-source SSL framework Lightly, which includes BarlowTwins , BYOL ,DCL, DCLW , DINO, Moco, NNCLR, SimCLR, SimSiam, SwaV.
> In our environment, its original baseline results are higher, and OR can further enhance these SSL methods.
> Please check the performance table and see the pdf material of the general response.
>
> >(W4)
>
> Thank you for your suggestions\! Our current results on Imagenet show that SO is able to improve more with small batchsize and small epochs.  Although the benefit of OR on ImageNet1k is smaller after increasing the epoch on ImageNet1k, it still improves transfer learning substantially.
> As for the results for larger epochs and larger batchsize, due to our limited time and computational resources,  the results of DINO and DINO (SO) on ImageNet with larger batchsize (2048) and epoch (300) are provided. In the follow-up work, we will definitely make up more epoch experiments to fully test the effect of OR.
> In addition, we added the results of various SSL methods on LIGHTLY framework cifar10, resnet18, 200 epochs, and 400 epochs, the OR can get the same improvement.
> Also, we report  BarlowTwins and BYOL results under Imagenet100 (ResNet 18, epoch 200, batch size 256).
> Please check the performance table in the general response.
>
> >(W5)
>
> By visualizing weight matrices and features, we can see that OR avoids the dimensional collapse of weights as well as features (Figure 3 in the paper), and it even likewise improves on feature whitening methods such as VICREG and BarlowTwins.  Please check the performance table in the general response.
>
> Without OR, the current SSL methods will not completely collapse as they introduced techniques such as using negative samples, self-distillation, clustering, and feature whitening. In this paper, we visualize the decaying eigenspace of weight matrices and features to observe the dimensional collapse, and we demonstrate the OR can effectively mitigate, if not prevent, the dimensional collapse.

---

> > ### Comment · Reviewer_sksr · 2024-08-13
> >
> > Here are some post-rebuttal comments regarding how the authors addressed my concerns:
> >
> > (A) Overall, the authors have satisfactorily addressed my concerns W1, W3, and W5.
> >
> > (B) Regarding W2: Demonstrating the benefits of adding the OR regularization to SSL methods is indeed a notable contribution. However, OR is applied directly, in a plug-and-play manner, without any modifications proposed in this work to "turn it into a plug-and-play method for SSL," as the authors stated. Nevertheless, I still consider the contribution of showing the advantages of OR in the context of SSL methods to be significant.
> >
> > (C) The authors provided many more experiments in the rebuttal, showing that OR consistently improves any baseline SSL method to which it is applied. As I mentioned in my initial review, this versatility of OR is the strongest aspect of this work!
> >
> > (D) However, it remains unclear to me whether OR could improve the performance of a well-tuned baseline. For instance, the reproduced DINO results with the ResNet50 backbone provided by the authors are still notably worse than the ResNet50 results reported in the DINO paper. This is the main remaining weakness of the paper.
> >
> > In conclusion, considering the above comments, I am going to increase my score. I am between "weak accept" and "borderline accept," leaning more towards "borderline accept" due to comment (D): it is unclear whether OR can enhance the performance of well-tuned baselines.

---

> ### Author Response · Authors · 2024-08-13
>
> Thank you for your kind response and suggestions.
>
> >(A)
>
> We are very happy that we have solved this part of your concerns and it has greatly improved the quality of our work.
>
> >(B)
>
> Indeed, as suggested by Reviewer KCGP, we intend in the future to make changes to the form of the OR to avoid the adjustment of the OR hyperparameters by the increase in model size.
>
> >(C)
>
> Thanks again for your recognition!
>
> >(D)
>
> DINO on Solo-learn results are really puzzling. For this reason, we performed DINO ResNet50 300 epoch experiments on the LIGHTLY framework. However, the results are still far from those in the original DINO article, and we found that this is due to the fact that **LIGHTLY does not perform multi-crop**, which has a fatal impact on the performance. Without multi-crop, as shown in the following table (results from the paper of DINO), we believe that LIGHTLY's results are consistent with the original DINO article.
>
> |  | KNN Top1 | Linear Top1 |
> | :---- | :---- | :---- |
> | DINO  (vit/s)  | 72.8 | 76.1 |
> | DINO (vit/s) w/o multi-crop | 67.9 | 72.5 |
> | DINO  (ResNet 50\) | 65.6 | 74.5 |
> | DINO (ResNet 50\) w/o multi-crop LIGHTLY | 59.8 | 71.9 |
> | DINO (ResNet 50\) w/o multi-crop  \+SO LIGHTLY | 60.4 | 72.2 |
>
> In the future, to be more convincing, we need to add multi-crop to LIGHTLY or conduct experiments in the DINO repository.

---

### Official Review · Reviewer_gNSa · 2024-07-12

**Soundness:** 4
**Presentation:** 4
**Contribution:** 4
**Rating:** 7
**Confidence:** 4

**Summary:**

This paper aims to improve self-supervised learning (SSL) by introducing orthogonality regularization (OR) during pretraining to prevent dimensional collapse. OR improses orthogonality constraint on weight matrices, not only preventing dimensional collapse of representation but also regularizing the weights from collapsing. The experimental results show that the benefit of the proposed method is consistent and widely applicable across datasets and architecture choices.

**Strengths:**

* This paper effectively communicates both the problem it addresses and the solution it presents. The motivation and solutions are compelling and presented in an easily understandable manner.
* The benefit of the proposed method appears to be consistent and significant.

**Weaknesses:**

* More discussion on the difference between the proposed orthogonal regularization methods would have been helpful. On the surface, the two methods are almost equally good.

**Questions:**

* In practice, a practitioner has to choose between SO and SRIP. What would be the authors' recommendation?
* Do you expect that applying dropout on the encoder's hidden neurons would result in a similar effect, as dropout is also meant to prevent feature co-adaptation? It is quite likely that the existing encoders have already employed dropout, so I wonder if dropout is insufficient to prevent the collapse.

---

> ### Author Rebuttal · Authors · 2024-08-06
>
> We appreciate your comments and feedback. In addition to the general response, we address your itemized concerns here.
>
> >More discussion on the difference between the proposed orthogonal regularization methods would have been helpful. On the surface, the two methods are almost equally good.
>
> Thanks for the advice, it helps a lot for those who intend to use OR\!
> Yes, both are almost the same in terms of effectiveness. SRIP is more time-consuming and takes up more memory than SO.
> So we chose SO for very time-consuming experiments (ImageNet and ImageNet100). Additionally, SRIP has theoretically **weaker (softer)** regularization constraints than SO because it uses the spectral norm instead of the L2 norm.
>
> Based on our experience with large-scale datasets and OR's results for the feature whitening method VICREG, we recommend SO. We add 10 SSL methods in the LIGHTLY framework, all of which use SO, and all of which obtain improved results. Please check the performance table in the general response.
>
> >Do you expect that applying dropout on the encoder's hidden neurons would result in a similar effect, as dropout is also meant to prevent feature co-adaptation? It is quite likely that the existing encoders have already employed dropout, so I wonder if dropout is insufficient to prevent the collapse.
>
> Thanks for raising this question. Indeed, dropout is also able to prevent feature co-adaptation. In SSL, there is no dropout in ResNet, but dropout is enabled in VITs. Our OR approach significantly improves the DINO with VITs (Table 2 in the paper), suggesting that single dropout cannot adequately prevent dimensional collapse at the three levels of the weight matrix, intermediate features, and representations.
> We can visualize Dropout's impact on all three in future work. Exploring the role of Dropout and OR in SSL is very interesting for further research.
>
> In fact, because of the widespread use of Dropout in the Transformer architecture, there has been some exploration of the impact of dropout on the features of SSL using the Transformer architecture. Dropout encourages the use of global features rather than local features in the Transformer architectures \[1\].
>
> \[1\] Luo, Jian, et al. "Dropout regularization for self-supervised learning of transformer encoder speech representation." *arXiv preprint arXiv:2107.04227* (2021).

---

> > ### Comment · Reviewer_gNSa · 2024-08-09
> > **Reply for rebuttal**
> >
> > Thank you for your response. Your answers clarified my questions. Other reviewers seem to have concerns regarding empirical results, of which I don't have a strong opinion, since I do not have much hands-on experience with SSL. In general, I believe the paper is well-structured and easy to understand. Hope the concerns regarding the experiments are resolved well during the discussion.

---

> > > ### Author Response · Authors · 2024-08-09
> > >
> > > Thank you very much for your kind reply and recognition of our work. Regarding empirical results, we also hope that we can address the concerns of other reviewers during the discussion phase.

---

### Official Review · Reviewer_KCGP · 2024-07-12

**Soundness:** 2
**Presentation:** 3
**Contribution:** 3
**Rating:** 5
**Confidence:** 4

**Summary:**

Dimensional collapse, characterized by the dominance of a few large eigenvalues, is a phenomenon in self-supervised learning (SSL) that can lead to redundant features and weight matrices. To address this, the authors propose orthogonal regularization (OR) during pretraining to promote orthogonality in weight matrices, rather than using common approaches that regularize features via the objective (e.g., Barlow Twins, VicReg). The authors conduct experiments with five SSL methods from the SoloLearn framework to demonstrate the benefit of OR.

**Strengths:**

The paper is well-written and provides a good introduction to the topic of representation collapse. The motivation for the approach is clearly stated, and the experimental setting is well documented. With small architectures like ResNet18, the effect of orthogonal regularization (OR) appears impressive at first glance. Although OR introduces some computational overhead, it is relatively small and may diminish further with larger batch sizes.

**Weaknesses:**

The soundness of some experiments seems doubtful. Results for WideResNet28x2 are consistently below those for ResNet18. Additionally, the results using Dino are generally poor, leading to all ViT experiments being below any ResNet baseline.

The performance of BYOL with OR, when trained for only 100 epochs with a batch size of 64, is even below the reported accuracy in SoloLearn. Considering the optimal training duration for BYOL is approximately 1000 epochs with a batch size of 4096, this comparison appears to be in a severely undertrained regime. Slightly longer training, 200 epochs with a batch size of 256, shows that the benefit of the OR approach diminishes rapidly.

Minor issue: Line 44 states, "We evaluate the effect of adding OR in seven modern SSL methods during pretraining," but results are shown for only five methods.

Additionally, since SSL approaches like Barlow Twins and VicReg aim to achieve orthogonality as well, there should be an experiment evaluating the effect of OR on these methods.

Using one learning rate (LR) for all methods in linear probing might not be optimal. It is cheap to attach multiple linear probing heads and find the most suitable LR for each method to ensure fair comparisons.

**Questions:**

Memory Overhead of OR: How much memory overhead does orthogonal regularization (OR) introduce?

Explanation for Lower Gamma in Larger Models: The paper uses a lower gamma for larger models, but why? One might think that larger models would require more regularization.

Results for Barlow Twins and VicReg: Why were results for Barlow Twins or VicReg not reported? Is the OR approach redundant when combined with these methods, which also aim to achieve orthogonality?

Balanced Eigenvalues and Performance Discrepancy: The eigenvalues of VicReg without a predictor (Appendix Fig 4) appear more balanced than those of BYOL OR (Fig 3), yet VicReg's performance is far worse.

**Limitations:**

The authors adequately addressed the limitations.

---

> ### Author Rebuttal · Authors · 2024-08-06
>
> We appreciate your comments and feedback. In addition to the general response, we address your itemized concerns here.
> > The soundness of some experiments
>
> Thanks for pointing out this. Solo-learn only provides the official config file (yaml) for resnet18 on cifar100, on which we replaced the backbone (VITs, ResNet50,WideResnet) and added the OR. To ensure a fair comparison, we did not modify any other hyperparameters. Replacing a backbone may require tuning these parameters to achieve the best results. For now, we just want to show that OR can be plug-and-play under all kinds of backbone.
> More importantly, we believe our OR method could boost the performance under different hyperparameter settings.
> Please check the performance table in the general response.
>
> >Additionally, the results using Dino
>
> In fact, VIT-small **outperforms** all the ResNet backbone results in DINO under the cifar100 experiment, please refer to Tables 1 and 2 in the paper.  Results of DINO in the ResNet series are poor, and we believe this is due to the parameters in the config files of its repository. For example, in DINO, its default config uses the lars optimizer, warmup\_teacher\_temperature\_epochs: 0, warmup\_start\_lr: 3e-05. However, the optimizer used for checkpoint results is SGD, lars=Ture, warmup\_teacher\_temperature\_epochs: 50, warmup\_start\_lr: 3e-03. These differences in hyperparameters cause our results to be different from their reported results. Our baseline and OR methods run entirely under their officially provided hyperparameter config.  To ensure fair comparisons, we did not modify the parameters in the original configs.
>
> >The performance of BYOL with OR
>
> We checked the parameters in its checkpoint and BYOL trained on ImageNet with a batch size of 128 instead of 64 in its yaml config file. We think this is the reason for the different results.
> Our current results on ImageNet1k show that SO can perform better with small batchsize and small epochs. Although the benefit of OR on ImageNet1k is smaller after increasing the epoch on ImageNet1k, it still improves transfer learning substantially.
>
> As for the results for larger epochs and larger batch sizes, the experiments of DINO and DINO (SO) on ImageNet with larger batchsize (2048) and epoch (300) are provided.In the follow-up work, we will conduct more experiments to fully test the effect of OR.
> **Please check the performance table in the general response.**
>
> >Minor issue:
>
> Sorry for our mistake. The SSL methods we have now include **6** methods implemented by solo-learn (MOCOv2plus, MOCOv3, DINO, NNBYOL, BYOL, VICREG) and **10** methods implemented by the LIGHTLY framework (BarlowTwins, BYOL, DCL, DCLW, DINO,  Moco, NNCLR, SimCLR, SimSiam, SwaV). Removing duplicates, there are in total **13** different SSL methods.
>
> >Additionally, since SSL
>
> Your suggestions are very constructive. We tried to add OR to VICREG and Barlow Twins (resnet18, cifar10). **Both gained a boost from OR**, suggesting that simply requiring the output of the projector to be whitened is not enough to avoid weight matrix collapse. We additionally visualized the weights of the former, please see the pdf material of the general response.
>
> >Using one learning rate (LR)
>
> Thanks for the suggestion, which is very reasonable. The representations produced by different methods may require different learning rates for the downstream tasks. We are only using the solo-learn settings for now. Considering the time constraints during the rebuttal period, we will try to adjust the learning rate later to get the best results.
> For fair comparisons, we added the results of 10 SSL methods using KNN classification under the LIGHTLY framework to visualize the quality of the representations better.
>
>
> >Memory Overhead of OR: How much memory overhead does orthogonal regularization (OR) introduce?
>
> For a specific backbone, OR traverses the parameters (linear and convolutional layers) in it.
> Let's assume that the largest W is a square matrix with shape n\*n, then the memory overhead needed is to compute W.T @ W which is also an n\*n matrix, and to introduce an identical matrix I, which is also an n\*n matrix, and storing the distance between the two requires an n\*n matrix. So in total, the memory overhead of the forward propagation is
> 4 \*n\*n\*(Floating Point Precision)/4.
> Using BarlowTwins in LIGHTLY with resnet18 as an example, the maximum memory usage without OR is 4.87GB, with SO it is 4.89GB. Therefore, the memory overhead of OR is quite low.
>
> >Explanation for Lower Gamma
>
> You are asking a very good question to promote understanding of OR loss.
> Because our OR is a penalized **summation of all parameters**, the larger the number of backbone parameters, the OR loss will increase rapidly, in order to be balanced with the SSL loss, we have only appropriately scaled down the Gmma by a multiplicity of 0.1.   We checked the SO loss of the randomly initialized ResNet18 and ResNet50 backbones (gamma= 1), which is 5873.4062 for the former and 217609.7031 for the latter, and the increase in the number of model parameters brings about a nearly **40-fold** increase in OR loss.
>
> >Balanced Eigenvalues and Performance Discrepancy
>
> Your notice is well worth further discussion. The whiteness (balanced eigenvalues) of representations and their performance do not grow in tandem, see Table 4 in \[1\].
> We visualized the original VICREG, and we can see that the original VICREG is much better than the version with the projector removed, but the whiteness degree of its representations is not as high as the one with the projector removed. Please see the pdf material of the general response.
>
> \[1\] He, Bobby, and Mete Ozay. "Exploring the gap between collapsed & whitened features in self-supervised learning." *International Conference on Machine Learning*. PMLR, 2022\.

---

> ### Comment · Reviewer_KCGP · 2024-08-10
> **Reply to the rebuttal**
>
> Thanks to the authors for the thorough response and the additional experiments.
>
> (A) Regarding experiments. It is not a fair comparison if one uses suboptimal hyperparameters for a baseline and then introduces a new hyperparameter that can be tuned. It does not provide a lot of insights of the usefulness of the method to a well tuned baseline. It only shows that the proposed regularization has some kind of effect on the training. The experiments with WideResnet and Dino in its current form are simply a distraction. Either the dataset is not large enough to apply a WideResNet or the hyperparameters do not work at all. For Dino, it is true that Dino ViT ourperforms Dino ResNet. But all other methods outperform the best Dino ViT result even with ResNet18. Therefore, the used hyperparameters or training schedule of the Dino setting do not work at all.
>
> (B) Memory overhead. Thank you, the fact that memory overhead is feasible and does not scale with the used batchsize is great.
>
> (C) Usefulness of the proposed method. One the one hand, the introduction of a new hyperparameter and the increased effort to tune it should lead to a significant increase in a well tuned baseline that is trained for sufficiently long time. The current work does not convince me that this is the case. On the other hand, the improvements with short training schedules and the improved transfer capabilities are noteworthy. To me this is very similar to the strengths and weaknesses of Barlow Twins (sensitive to the number of features in the projector space). The inductive bias of the independence regularization speeds up learning and creates a representation that is robust to shifts, but can also be detrimental to fit a function in domain, given enough samples and training time. A study how to the gamma scales with model size and training schedule that might reduce the necessary tuning of the introduced hyperparameter would be of interest. Futhermore a direct comparison to BarlowTwins/VicReg to specifically measure the benefits of improved efficiency regarding training time or training data as well as robustness to transfer learning could improve the work in my opinion.
>
> I will keep my score, but I am not opposed to the acceptance of the work given the predominantly positive scores by other reviewers. EDIT: In expectation that the authors incorporate the feedback into their final version, I will raise my score.

---

> > ### Author Response · Authors · 2024-08-11
> >
> > Thank you for your response and further suggestions.
> >
> > > DINO results on solo-learn
> >
> > Indeed, all of our methods follow the config file provided by solo-learn, where the hyperparameters are inconsistent with the checkpoints, leading to rather puzzling results for DINO on solo-learn, which is why we supplemented our experiments with the performance of DINO on other frameworks. **Our additional experiments show that DINO on CIFAR10 and ImageNet1k under the LIGHTLY framework is more in line with expectations, and OR improves their results**. We will next replace the backbones (WideResNet and VITs) in the LIGHTLY framework.
> >
> > >  Memory overhead
> >
> > Yes, OR brings only a very small memory overhead and is not affected by batch size, and is one of its major advantages over other feature whitening based methods, we will add these discussions to the manuscript!
> >
> > >  One the one hand, the introduction of a new hyperparameter and the increased effort to tune it should lead to a significant increase in a well tuned baseline that is trained for sufficiently long time. The current work does not convince me that this is the case. On the other hand, the improvements with short training schedules and the improved transfer capabilities are noteworthy
> >
> > Due to limited time and GPU resources, we only tested the effect of OR in DINO 300 epochs on ImageNet1K during the rebuttal period, more epochs of the experiment are catching up to run, and your suggestion is very reasonable. We will not only test the performance under the original dataset, but its performance in transfer learning will also be reported, considering that we experimentally found that although the improvement is not significant under the original data, OR can still bring significant improvement in the transfer learning scenario.
> >
> > >   A study how to the gamma scales with model size and training schedule that might reduce the necessary tuning of the introduced hyperparameter would be of interest.
> >
> > We totally agree with you that less tuning of the OR hyperparameters can make the OR more plug-and-play. In the future, we will try not to add up the OR loss for each weight matrix, but average it. This way, the model size won't affect the magnitude of OR loss much.
> >
> > > BarlowTwins/VicReg to specifically measure the benefits of improved efficiency regarding training time or training data as well as robustness to transfer learning could improve the work
> >
> > Thanks again for your kind suggestions. In additional experiments, BarlowTwins gained boosts from OR on both CIFAR10 and ImageNet100. We will test the results of BarlowTwins and VICREG on ImageNet1K further with more training epochs and transfer learning scenarios.

---

> ### Comment · Reviewer_KCGP · 2024-08-12
>
> My point was that consistent results do not directly render an experiment useful, as there is too much room for improvement when the tuning is not done correctly.
>
> I want to clarify that I did not want to critique the new experiments, which are small scale, but look solid and cover a lot of methods.

---

> > ### Author Response · Authors · 2024-08-12
> >
> > Thanks again for your suggestion and reply.
> > Our next experiments with backbone replacement will try to adjust the hyperparameters of the corresponding SSL to optimize the original method and then add the OR. This setup will definitely be more convincing.

---

### Author Rebuttal · Authors · 2024-08-06

We thank all reviewers for their questions and constructive feedback.   Here, we respond to the five core issues of common interest:

**Solo-learning baseline settings**
The baseline results of our report are the results of the newest official config file, which are different from the checkpoint results.  **This is because the configs in the checkpoint itself are different from the newest official config file**. For example, in DINO, its default config uses the lars optimizer, warmup\_teacher\_temperature\_epochs: 0, warmup\_start\_lr: 3e-05. However, the optimizer used for checkpoint results is SGD, lars=Ture, warmup\_teacher\_temperature\_epochs: 50, warmup\_start\_lr: 3e-03.

**More SSL methods under different hyperparameter settings**
The SSL methods we have now include **6** methods implemented by solo-learn (MOCOv2plus, MOCOv3, DINO, NNBYOL, BYOL, VICREG) and **10** methods implemented by the LIGHTLY framework (BarlowTwins, BYOL, DCL, DCLW, DINO,  Moco, NNCLR, SimCLR, SimSiam, SwaV). Removing duplicates, there are in total **13** different SSL methods.
Under the LIGHTLY framework, we tested the effect of SO under CIFAR10, ImageNet100, and ImageNet1K.

|  KNN Top1 (CIFAR10) | Batch 512 Epoch 200 |  |  | Batch 512 Epoch 400 |  |
| :---- | :---- | :---- | :---- | :---- | :---- |
|  |  Report |  Reproduced |  w/ SO |  Reproduced |  w/ SO |
| Barlow Twins | 81.9 | 83.58 | **84.78** | 85.91 | **86.25** |
| BYOL | 86.8 | 86.94 | **87.01** | 89.64 | **90.02** |
| DCL | 84.0 | 83.38 | **84.10** | 85.95 | **86.27** |
| DCLW | 82.4 | 82.42 | **82.73** | 85.25 | **85.71** |
| DINO | 81.3 | 81.87 | **81.95** | \\ | \\ |
| Moco | 84.7 | 85.19 | **85.32** | \\ | \\ |
| NNCLR | 81.5 | 82.31 | **82.34** | \\ | \\ |
| SimCLR | 84.8 | 84.55 | **84.84** | \\ | \\ |
| SimSiam | 76.4 | 79.27 | **84.31** | \\ | \\ |
| SwaV | 84.2 | 83.10 | **83.67** | \\ | \\ |

we can see that even under different implementations with different hyperparameter settings (in our environment, our reproduced results are even higher), OR can still have a stable improvement in SSL performance.

**Relationships with feature whitening**
Feature whitening methods like Barlow Twins and VICREG aim to avoid dimensional collapse of intermediate features as well as weights by whitening the output of the projector.
Our OR, on the other hand, acts directly on the weight matrix to indirectly avoid the dimensional collapse of features.

Thanks for the suggestions of Reviewer KCGP, sksr, Yvo9. We supplement the experiments based on feature whitening methods such as Barlow Twins (LIGHTLY) and VICREG (Solo-learn), **OR can further improve their performance**. We also visualize weights and features of  BYOL w/ vic, VICREG, and VICREG w/ OR (**please see the pdf material**), which are not able to avoid dimensional collapse of their weight matrices by feature whitening alone.
Simply adding feature whitening to the original SSL method will even degrade its performance, in contrast, our plug-and-play OR method can improve the performance.
| vicreg w/o Projector | vicreg | vicreg \+ SO (1e-6) | vicreg \+ SRIP (1e-3) | BOYL | BYOL+ SO (1e-6) | BYOL+ vic (1e-3) | BYOL+ vic (1e-4) | BYOL+ vic (1e-5) |
| :---- | :---- | :---- | :---- | :---- | :---- | :---- | :---- | :---- |
| 88.64 | 91.64 | **92.35** | 92.22 | 92.92 | **93.04** | 92.35 | 92.38 | 92.66 |
| 99.57 | 99.73 | **99.79** | 99.67 | 99.83 | **99.84** | 99.85 | 99.82 | 99.79 |

**Contributions**
From our observations, although existing methods such as VICREG deal with the collapse of features via whitening features, they can not prevent the dimensional collapse of the weight matrices. We are not only the first work to systematically study dimensional collapse at all levels in SSL but also provide new solutions to address the dimensional collapse by constraining the weight matrix.

In a broader sense, OR is just one of the methods we use to constrain the weight matrix in SSL, as it is intuitive to avoid the collapse of the weight matrix and has deterministic effects on the features. We only employed the widely used form of OR, but we wanted to share it with the community given its plug-and-play and interpretability in SSL methods.

**Larger datasets with larger settings**
We understand that examining the effect of OR under larger batchsize, more epochs, especially the large-scale dataset ImageNet is necessary.
Due to our limited time and computational resources,  We have to choose to complete the following experiments.
1. ImageNet100, Barrow Twins and BYOL in the LIGHTLY framework (ResNet18 epoch 200 batchsize 256\)

|  | Barrow Twins w/SO | Barrow Twins  (Reproduced) | Barrow Twins   | BYOL w/SO | BYOL  (Reproduced)  | BYOL   |
| :---- | :---- | :---- | :---- | :---- | :---- | :---- |
| KNN Top1 | **57.0** | 56.6 | 46.5 | **52.1** | 51.7 | 43.9 |

2. ImageNet1k, DINO in the LIGHTLY framework (ResNet50 epoch 300 batchsize 2048\)

|  | DINO w/SO | DINO (Reproduced) |
| :---- | :---- | :---- |
| KNN Top1 | **60.40** | 59.77 |
| KNN Top5 | **85.80** | 85.38 |
| Linear Top1 | **72.18** | 71.88 |
| Linear Top5 | **90.47** | 90.2 |

3. ImageNet1k, BYOL in the Solo-learn framework:  Transfer Learning

|  | ImageNet Top1 | Flowers102 Top1  |
| :---- | :---- | :---- |
| Batchsize 64  Epoch 100 | 65.81 | 73.817 |
| w/ SO | **67.84** | **78.956** |
| Btachsize 256 Epoch 200 | 69.76 | 76.452 |
| w/ SO | **70.16**  | **80.371** |

Although the benefit of OR on ImageNet1k is smaller after increasing the epoch on ImageNet1k, it still improves transfer learning substantially.

Overall, we would like to thank the reviewers once again for your detailed and effective suggestions. Improving and supplementing the experiments based on your suggestions will undoubtedly make our article more convincing.

---

### Decision · Program_Chairs · 2024-09-25

**Decision:**

Accept (poster)

**Comment:**

**Summary of the paper:**
The paper proposes weight regularization in order to avoid feature dimensionality collapse in self-supervised learning. It reguralizes the weights by adding a loss function, of two possible types, that encourage orthogonality of the weight matrix as well as a uniform spectral domain. It shows on several architectures, a few SSL algorithms, and a couple of datasets, that the addition of their loss often improves the downstream performance.

**Summary of the reviews:**
The reviewers found the paper well-written, and the proposed method simple but well-motivated. They mostly found the results convincing in that the proposed loss function usually shows benefits over its absence.

On the other hand, the reviewers had doubts about the soundness of the experimental setup when in comes to the paper’s results on the established methods and architectures. They also pointed a lack of comparison with other methods to avoid collapse such as Barlow Twins and VIC regularization. Two reviewers found the technical contribution to be low.

**Summary of the rebuttal and discussions:**
The authors point out the subpar results is due to the plugging of architectures into their used codebase without hyperparameter optimization or, in the case of DINO, due to the lack of the essential multicrop augmentation. They further offered new experiments by adding their loss function on top of VICreg and BT.

**Consolidation report:**

During the discussion the reviewers were mostly (but not fully) convinced regarding the occasional subpar results of the baselines, especially given that the authors promised to improve and reflect on these discrepancies in the final version. As a result they all moved their ratings towards acceptance.

Dimensionality collapse is an important issue with SSL which is an important field, this can compensate for the lack of general technical contribution pointed out by reviewers

**Recommendation:**
All in all, the AC agrees with the unanimous view of the reviewers and recommends acceptance.

In the final version, it is important to,
- as promised, improve and reflect upon the reported results as was partially done during the rebuttal.
- cite the following [a] paper that is published at NeurIPS 2023 workshops and is directly related to the proposed method.

[a] “WERank: Rank Degradation Prevention for Self-Supervised Learning via Weight Regularization” NeurIPS 2023 workshop on Self-Supervised Learning: Theory and Practice